# The ISIMIP Groundwater Sector: A Framework for Ensemble Modeling of Global Change Impacts on Groundwater

Robert Reinecke[1], Tanjila Akhter[2], Annemarie Bäthge[1], Ricarda Dietrich[1], Sebastian Gnann[3], Simon N. Gosling[4], Danielle Grogan[5], Andreas Hartmann[6], Stefan Kollet[7], Rohini Kumar[8], Richard Lammers[5], Sida Liu[9], Yan Liu[7], Nils Moosdorf[10,11], Bibi Naz[7], Sara Nazari[10], Chibuike Orazulike[6], Yadu Pokhrel[2], Jacob Schewe[12], Mikhail Smilovic[13,14], Maryna Strokal[8], Wim Thiery[15], Yoshihide Wada[16], Shan Zuidema[4], Inge de Graaf[8]

*Correspondence to*: Robert Reinecke (reinecke@uni-mainz.de)

[1]Institute of Geography, Johannes Gutenberg-University, Mainz, Mainz, Germany

[2]Department of Civil and Environmental Engineering, Michigan State University, East Lansing, Michigan, USA

[3]Chair of Hydrology, University of Freiburg, Freiburg, Germany

[4]School of Geography, University of Nottingham, Nottingham, United Kingdom

[5]Institute for the Study of Earth, Oceans, and Space, University of New Hapshire, Durham, New Hampshire, USA

[6]Institute of Groundwater Management, TU Dresden, Dresden, Germany

[7]Agrosphere (IBG-3), Institute for Bio- and Geosciences, Research Centre Juelich, Juelich, Germany

[8]Department Computational Hydrosystems, Helmholtz Centre for Environmental Research GmbH - UFZ, Leipzig

[9]Earth Systems and Global Change Group, Wageningen University and Research, Wageningen, the Netherlands

[10]Leibniz Centre for Tropical Marine Research (ZMT), Bremen, Germany

[11]Institute of Geosciences, Kiel University, Kiel, Germany

[12]Potsdam Institute for Climate Impact Research, Member of the Leibniz Association, Potsdam, Germany

[13]Water Security Research Group, International Institute for Applied Systems Analysis, Laxenburg, Austria

[14]Chair of Hydrology and Water Resources, ETH Zurich, Zürich, Switzerland

[15]Vrije Universiteit Brussel, Department of Water and Climate, Brussels, Belgium

[16]Biological and Environmental Science and Engineering Division, King Abdullah University of Science and Technology, Thuwal, Saudi Arabia

**Abstract**

Groundwater serves as a crucial freshwater resource for people and ecosystems, playing a vital role in adapting to climate change. Yet, its availability and dynamics are affected by climate variations, changes in land use, and abstraction. Despite its importance, our understanding of how global change will influence groundwater in the future remains limited. Multi-model ensembles are powerful tools for impact assessments; compared to single-model studies, they provide a more comprehensive understanding of uncertainties and enhance the robustness of projections by capturing a range of possible outcomes. However, to date, no ensemble of groundwater models has been available to assess the impacts of global change. Here, we present the new Groundwater sector within ISIMIP, which combines multiple global, continental, and regional-scale groundwater models. We describe the rationale for the sector, the sectoral output variables that underpinned the modeling protocol, and showcase current model differences and possible future analysis. Currently, eight models are participating in this sector, ranging

from gradient-based groundwater models to specialized karst recharge models, each producing up to 19 out of 23 modeling protocol-defined output variables. To showcase the benefits of a joint sector, we utilize available model outputs of the participating models to show the substantial differences in estimating water table depth (global arithmetic mean 6 - 127 m) and groundwater recharge (global arithmetic mean 78 - 228 mm/y), which is consistent with recent studies on the uncertainty of groundwater models, but with distinct spatial patterns. We further outline synergies with 13 of the 17 existing ISIMIP sectors and specifically discuss those with the global water and water quality sectors. Finally, this paper outlines a vision for ensemble-based groundwater studies that can contribute to a better understanding of the impacts of climate change, land use change, environmental change, and socio-economic change on the world's largest accessible freshwater store – groundwater.

**1 Introduction**

Groundwater is the world's largest accessible freshwater resource, vital for human and environmental well-being (Huggins et al., 2023; Scanlon et al., 2023), serving as a critical buffer against water scarcity and surface water pollution (Foster and Chilton, 2003; Schwartz and Ibaraki, 2011). It supports irrigated agriculture, which supports 17% of global cropland and 40% of food production (Döll and Siebert, 2002; Perez et al., 2024; United Nations, 2022; Rodella et al., 2023). However, unsustainable extraction in many regions has led to declining groundwater levels, the drying of rivers, lakes and wells, land subsidence, seawater intrusion, and aquifer depletion (e.g., Bierkens and Wada (2019); de Graaf et al. (2019); Rodell et al. (2009)).

The pressure on groundwater systems intensifies due to the combined effects of population growth, socioeconomic development, agricultural intensification (Niazi et al. 2024; Wada et al. 2012), and climate change (Taylor et al., 2013; Gleeson et al., 2020, Cuthbert et al., 2023, Huggins et al., 2023), e.g., through a change in groundwater recharge (Portmann et al., 2013; Hartmann et al. 2017; Reinecke et al., 2021; Berghuijs et al., 2024; Kumar et al. 2025). Rising temperatures and altered precipitation patterns are already reshaping water availability and demand, with significant implications for groundwater use. For instance, changing aridity is expected to influence groundwater recharge rates (Berghuijs et al., 2024), yet the consequences for groundwater level dynamics remain unclear (Moeck et al., 2024; Cuthbert et al., 2019), and how possible changes will affect groundwater's role in sustaining ecosystems, agriculture, and human water supplies.

Understanding the impacts of climate change and the globalized socio-economy on groundwater systems (Rodella et al., 2023; Gisser et al., 1980) requires a large-scale perspective that extends from continental to global scales (Haqiqi et al., 2023; Konar et al., 2013; Dalin et al., 2017, Gleeson et al., 2021). While groundwater management is traditionally conducted at local or regional scales (Gleeson et al., 2014), aquifers often span administrative boundaries, and overextraction in one area can have far-reaching effects not captured by a local model. Moreover, groundwater plays a critical role in the global hydrological cycle, influencing surface energy distribution, soil moisture, and evapotranspiration through processes such as capillary rise (Condon and Maxwell, 2019; Maxwell et al., 2016) and supplying surface waters with baseflow (Winter, 2007; Xie et al., 2024). These interactions underscore the importance of groundwater in buffering climate dynamics over extended temporal and spatial scales (Keune et al., 2018) and underscore the need for a global perspective of the water-climate cycle. While large-scale climate-groundwater interactions are starting to become understood (Cuthbert et al., 2019), current

global water and climate models may not always capture these feedbacks as most either do not consider
groundwater at all or only include a simplified storage bucket, limiting our understanding of how climate change
will affect the water cycle as a whole (Gleeson et al., 2021; Condon et al. 2021).
The inclusion of groundwater dynamics in global hydrological models remains a considerable challenge due to
data limitations and computational demands (Gleeson et al., 2021). Simplified representations, e.g., linear
reservoir (Telteu et al., 2021), often fail to capture the complexity of groundwater-surface water interactions,
lateral flows at local or regional scales, or the feedback between groundwater pumping and streamflow (de Graaf
et al., 2017; Reinecke et al., 2019). These processes are crucial for evaluating water availability, particularly in
regions heavily dependent on groundwater. For instance, lateral flows sustain downstream river baseflows and
groundwater availability, which, in turn, impact water quality and ecological health (Schaller and Fan, 2009; Liu
et al., 2020). Not including head dynamics may lead to overestimation of groundwater depletion (Bierkens and
Wada, 2019). Multiple continental to global-scale groundwater models have been developed in recent years to
represent these critical processes (for an overview, see also Condon et al. (2021) and Gleeson et al. (2021).
While current model ensembles of global water assessments have not yet incorporated gradient-based
groundwater processes, they have already significantly advanced our understanding of the large-scale
groundwater system. The Inter-Sectoral Impact Model Intercomparison Project (ISIMIP), analogous to the
Coupled Model Intercomparison Project (CMIP) for climate models (Eyring et al., 2016a), is a well-established
community project to carry out model ensemble experiments for climate impact assessments (Frieler et al., 2017;
2024; 2025). The current generation of models in the Global Water Sector of ISIMIP often represents groundwater
as a simplified storage that receives recharge, releases baseflow, and can be pumped (Telteu et al., 2021). Still, it
lacks lateral connectivity and head-based surface-groundwater fluxes. Nevertheless, the ISIMIP water sector
provided important insights on, for example, future changes and hotspots in global terrestrial water storage
(Pokhrel et al., 2021), environmental flows (Thompson et al., 2021), the planetary boundary for freshwater change
(Porkka et al., 2024), uncertainties in the calculation of groundwater recharge (Reinecke et al., 2021), and the
development of methodological frameworks to compare model ensembles (Gnann et al., 2023).
Here, we present a new sector in ISIMIP called the ISIMIP Groundwater Sector, which integrates models of the
groundwater community that operate at regional (at least multiple km² (Gleeson and Paszkowski, 2014)) to global
scales and are committed to providing model simulations to this new sector. The Groundwater sector aims to
provide a comprehensive understanding of the current state of groundwater representation in large-scale models,
identify groundwater-related uncertainties, enhance the robustness of predictions regarding the impact of global
change on groundwater and connected systems through model ensembles, and provide insight into how to most
reliably and efficiently model groundwater on regional to global scales. The new Groundwater sector is a separate
but complementary sector to the existing Global Water sector. To our knowledge, there are currently no long-term
community efforts for a structured model intercomparison project for groundwater models. While studies have
benchmarked different modeling approaches (e.g., Maxwell et al. 2014), compared model outputs (Reinecke et
al., 2021; 2024), or collected information on where and how we model groundwater (Telteu et al., 2021; Zipper
et al., 2023; Zamrsky et al., 2025), no effort yet aims at forcing different groundwater models with the same
climate and human forcings for different scenarios.
Specifically, the ISIMIP Groundwater sector will compile a model ensemble that enables us to assess the impact
of global change on various groundwater-related variables and quantify model and scenario-related uncertainties.
These insights can then be used to quantify the impacts of global change on, for example, water availability and
in relation to other sectors impacted by changes in groundwater. The new sector welcomes all models that are
relevant to assessing the impacts of global change on groundwater-related variables. While the current set of
models presented here focuses on different physical representations of groundwater, future developments could
also include models that account for hydro-economic aspects of groundwater (e.g., Niazi et al. 2025; Kahil et al.
2025). The ISIMIP Groundwater sector has natural linkages with other ISIMIP sectors, such as Global Water,
Water Quality, Regional Water, and Agriculture. This paper will highlight the connections between groundwater
and different ISIMIP sectors, providing an opportunity to enhance our understanding of how modeling choices
affect groundwater simulation dynamics.
In this manuscript, we provide an overview of the current ISIMIP framework with an emphasis on how the new
sector is embedded in the current project in Section 2. The current generation of groundwater models participating
in this effort is described and compared, and we define a list of output variables that form the foundation of the
sector's model intercomparison protocol in Section 3. In 4, we showcase current model differences and possible
future analysis. The connections to other sectors are discussed in Section 5, and Section 6 provides an outlook on
future scientific goals for the groundwater sector.

**2 The ISIMIP framework**
ISIMIP aims to provide a framework for consistent climate impact data across sectors and scales. It facilitates
model evaluation and improvement, enables climate change impact assessments across sectors, and provides
robust projections of climate change impacts under different socioeconomic scenarios. ISIMIP uses a subset of
bias-adjusted climate models from the CMIP6 ensemble. The subset is selected to represent the broader CMIP6
ensemble while maintaining computational feasibility for impact studies (Lange, 2021).
ISIMIP has undergone multiple phases, with the current phase being ISIMIP3. The simulation rounds consist of
two main components: ISIMIP3a and ISIMIP3b, each serving distinct purposes. ISIMIP3a focuses on model
evaluation and the attribution of observed climate impacts, covering the historical period up to 2021. It utilizes
observational climate and socioeconomic data and includes a counterfactual "no climate change baseline" using
detrended climate data for impact attribution. Additionally, ISIMIP3a includes sensitivity experiments with high-
resolution historical climate forcing and water management sensitivity experiments. In contrast, ISIMIP3b aims
to quantify climate-related risks under various future scenarios, covering pre-industrial, historical, and future
projections. ISIMIP3b is divided into three groups: Group I for pre-industrial and historical periods, Group II for
future projections with fixed 2015 direct human forcing, and Group III for future projections with changing
socioeconomic conditions and representation of adaptation. Despite their differences in focus, time periods, and
data sources, both ISIMIP3a and ISIMIP3b require the use of the same impact model version to ensure consistent
interpretation of output data, thereby contributing to ISIMIP's overall goal of providing a framework for consistent
climate impact data across sectors and scales.

The creation of a new ISIMIP Groundwater sector is not linked to any funding and is a community-driven effort that includes all modeling groups that wish to participate. During the creation process, multiple groups and institutions were contacted to participate, and additional modeling groups are welcome to join the sector in the future. Models participating in the sectors do not need to be able to model all variables and scenarios defined in the protocol. ISIMIP sectors can be linked to broader thematic concepts, such as Agriculture, or can focus on specific components of the Earth system, such as Lakes or Groundwater (see also https://www.isimip.org/about/#sectors-and-contacts). The separation into these sectors is driven by the availability of models that can be integrated into a model-intercomparison framework, which is based on the same climatic and human forcings and produces a set of comparable output variables. We would like to note that groundwater is not an isolated system, but rather part of the water cycle and the Earth system as a whole. Focusing on it within a dedicated sector aligns well with the existing models and is useful for studying groundwater systems in a thematically focused way. Collaboration (and perhaps integration) with sectors like the Global Water sector is possible and desirable in the future. The global water sector focuses on using the ISIMIP protocol to drive a diverse set of global water models (including hydrological and land surface models; Reinecke et al. 2025b) and to produce output variables that capture diverse hydrologic processes, such as discharge, as well as human water use. We discuss possible future synergies with other existing ISIMIP sectors in Section 5.

In the short term, the Groundwater sector will focus on the historical period from 1901 to 2019 in ISIMIP3a (https://protocol.isimip.org/#/ISIMIP3a/water_global/groundwater), using climate-related forcing based on observational data (obsclim) and the direct human forcing based on historical data (histsoc). We aim to use these simulations for an in-depth model comparison, including a comparison to observational data such as time series of water table depth (e.g., Jasechko et al., 2024) and by utilizing so-called functional relationships (Reinecke et al., 2024; Gnann et al., 2023). Functional relationships can be defined as covariations of variables across space and/or time, and they are a key aspect of our theoretical knowledge of Earth's functioning. Examples include relationships between precipitation and groundwater recharge (Gnann et al. 2023; Berghuijs et al. 2024) or between topographic slope and water table depth (Reinecke et al., 2024).

Carrying out the ISIMIP experiments in the groundwater sector will yield a new understanding of how these models differ, why they differ, and how they could be improved. These experiments will further help to disentangle the impacts of climate change and water management, specifically through ensemble runs of future scenarios using ISIMIP3 inputs.

## 3 The current generation of groundwater models in the sector

Many large-scale groundwater models are already participating in the sector (Table 1), and we expect it to expand further. The current models are mainly global-scale, with some having a particular regional focus, and primarily using daily timesteps.

While the primary modeling purpose of most models is to simulate parts of the terrestrial water cycle, they all focus on different aspects (such as karst recharge or seawater intrusion), most investigate interactions between groundwater and land surface processes, and account for human water uses. Two models (V2KARST and GGR) have distinct purposes in modeling groundwater recharge and do not model any head-based groundwater fluxes.

Conceptually, the models may be classified according to Condon et al. (2021) into five categories: (a) lumped
models with static groundwater configurations of long-term mass balance, (b) saturated groundwater flow with
recharge, and surface water exchange fluxes as upper boundary conditions without lateral fluxes, (c) quasi 3D
models with variably saturated flow in the soil column and a dynamic water table as a lower boundary condition,
(d) saturated flow models solving mainly the Darcy equation, (e) and variably saturated flow which is calculated
as three-dimensional flow throughout the entire subsurface below and above the water table. See Condon et al.
(2021) and also Gleeson et al. (2021) for a more detailed overview and discussion of approaches. Half of the
models (Table 1) simulate a saturated subsurface flux (d), while V2KARST and GGR mainly use a 1D vertical
approach (b), and others simulate a combination of multiple approaches (ParFlow, Table 1) or can switch between
different approaches (CWatM, Table 1).
The sector protocol is defined at https://protocol.isimip.org/#/ISIMIP3a/groundwater and will be updated over
time. We have defined multiple joint outputs for this sector (23 variables in total), but not all models can yet
provide all outputs (Table 2). Models can provide 1-19 outputs (11 on average), and multiple models have
additional outputs that are currently under development. The global water sector also contains groundwater-related
variables (Table A2), enabling groundwater-related analysis. We list them here to show their close connection to
the global water sector and facilitate an overview of future groundwater-related studies.
The current sector protocol defines a targeted spatial resolution of 5 arcmin, as this represents not only the
resolution achievable by most global models but also the coarsest resolution at which meaningful representation
of groundwater dynamics, particularly lateral groundwater flows and water table depths, can still be captured
(Gleeson et al., 2021). ISIMIP3 also specifies experiments with different spatial resolutions, but whether this is
achievable with a sub-ensemble of the presented models remains unclear, as it depends on the available
computational time, flexibility of model setups, and data availability. To ensure consistency and comparability,
the model outputs are currently post-processed by the modeling groups to aggregate their outputs to the protocol-
specified spatial and temporal resolutions.
**Table 1**: Summary of all models participating in the ISIMIP Groundwater sector. This table lists only models that
add new variables to the ISIMIP protocol. Models already part of the global water sector and providing other
groundwater-related variables are not listed here. (GMD discussion formatting requires a portrait instead of a
landscape table)

| Model name | Main model purpose | Coupling with other models | Spatial domain and resolution | Temporal resolution | Hydrogeological configuration, e.g. number of layers | Conceptual model according to Condon et al. | Calibrated | Representation of groundwater use | Main Reference |
|---|---|---|---|---|---|---|---|---|---|
| Water Balance Model (WBM) | Representation of the terrestrial hydrologic cycle, including | - | Global and regional. Spatial resolution defined by the input | Sub-daily, Daily, Multi-day | 1 soil layer, 2 groundwater layers | d. | Globally: no, regional: yes (NE, US) | Through calculated abstractions from groundwater. | Grogan et al. (2022) With groundwater methods based on |

| | | | | | | | | | |
|---|---|---|---|---|---|---|---|---|---|
| | human interactions. | | river network. | | | | | | de Graaf et al. (2015); de Graaf et al. (2017). |
| Community Land Model (CLM) | To simulate surface and sub-surface hydrologic processes, including crop growth, irrigation, and groundwater withdrawal. | Community Earth System Model (CESM) | Global and regional (0.05 (regional), 0.1, 0.25, and 0.5 degree (global)) | Sub-Daily | 20 soil layers extending up to 8.5 m; 1 aquifer layer, unconfined | c. | No | Yes | Felfelani et al. (2021); Lawrence et al. (2019); Akhter et al. (2025) |
| Community Water Model (CWatM) | To reproduce main hydrologic processes, including water management on regional to global scales. | MODFLOW (optional) | Global, regional, subbasin (30 arcseconds, 1 km, 1 arc-min, 5 arc-min, 30 arc-min) | Daily | Standard: 1 with MODFLOW: variable | Standard: a./b. With MODFLOW: d. | Globally: yes (with discharge), regional: tailored | Yes | Guillaumot et al. (2022); Burek et al. (2020) |
| Global Gradient-based Groundwater Model (G³M) | Understanding of surface water, coastal, and ecosystem interaction with groundwater. | WaterGAP (Müller Schmied et al., 2016) | Global (5 arc minutes) | Daily, monthly, or yearly | 2 layers, second layer with a reduced hydraulic conductivity | d. | No | Through calculated net abstractions from groundwater of WaterGAP | Reinecke et al. (2019); Kretschmer et al. (2025) |
| VIC-WUR-MODFLO | Grid-based macro- | WOFOST (WOrld FOod | Regionally and globally: | Sub-daily to monthly | 3 soil layers (variable | d. | Globally: no, | Through calculated demands | Liu et al in prep.; |

| Model | Description | Base Model | Resolution | Time step | Layers | | Groundwater | Allocation | Reference |
|---|---|---|---|---|---|---|---|---|---|
| W (VIC-wur) | scale hydrological model that solves both the surface energy balance and water balance equations. | STudies) (Droppers et al 2021) | 5 arcminutes | | thickness), 2 groundwater layers (variable thickness, confined/unconfined systems. | | regional: yes | and allocation to surface water/ groundwater. | Droppers et al. 2020.; Liang et al. (1994) |
| V2KARST | A grid-based vegetation–recharge model for the global karst areas. | - | Globally: 0.25 arc degree | Daily | three soil layers and one epikarst layer | b. | Yes, based on global karst landscapes | no | Sarrazin et al. (2018) |
| Global Groundwater Rain-fed Recharge (GGR) | A grid-based three-layer water balance model to estimate the daily global rain-fed groundwater recharge | - | 180.0°W to 180.0°E longitudes and 60.0°N to 60.0°S latitudes, 0.1 degree | Daily | 2 soil layers and 1 groundwater layer of variable thickness | b. | No | No | Nazari et al. (2025) |
| ParFlow | 3D continuum simulations of variably saturated groundwater-surface water and land surface processes. | Common Land Model, CLM (Maxwell and Miller, 2005; Kollet and Maxwell, 2008), Terrestrial Systems Modeling Platform | Regionally and globally, $10^0 - 10^1$km | Variable | Variable | a. - e. | Yes, in engineering applications | Yes | Kuffour et al. (2020) |

| | | (Gasper et al., 2014), WRF (Maxwell et al., 2011) | | | | | | | | |
|---|---|---|---|---|---|---|---|---|---|---|








**Table 2:** List of output variables in the ISIMIP3a Groundwater sector. The spatial resolution is five arcminutes (even if some models simulate at a higher or coarser resolution), and the temporal resolution is monthly. Most models also simulate daily timesteps, but as most groundwater movement happens across longer time scales, we unified the unit to months. A "*" indicates that a model is able to produce the necessary output. A "+" indicates that this output is currently under development. (GMD discussion formatting requires a portrait instead of a landscape table)

| Groundwater sector output variables | | Unit | WBM | CLM | CWatM | G³M | VIC-wur | V2KARST | GGR | ParFlow |
|---|---|---|---|---|---|---|---|---|---|---|
| Name | Description | | | | | | | | | |
| Capillary rise | Upward flux from groundwater to soil (leaving aquifer = negative value). | m3 m-2 month-1 | | * | * | | * | | | * |
| Diffuse groundwater recharge | Downwards flux from soil to groundwater (entering aquifer = positive value). The unit kg m$^{-2}$ s$^{-1}$ is equal to mm s$^{-1}$. Unit is kept equal to the global water sector. | kg m-2 s-1 | * | * | * | | * | * | * | * |
| Groundwater abstractions | Groundwater pumped from the aquifer. | m3 m-2 month-1 | * | * | * | | + | | + | |
| Groundwater abstractions (domestic) | *Groundwater abstractions* that are intended for domestic water use. | m3 m-2 month-1 | * | | * | | + | | + | |

| | | | | | | | | | |
|---|---|---|---|---|---|---|---|---|---|
| Groundwater abstractions (industries) | *Groundwater abstractions* that are intended for industrial water use. | m3 m-2 month-1 | * | | * | | + | + | |
| Groundwater abstractions (irrigation) | *Groundwater abstractions* that are intended for irrigational water use. | m3 m-2 month-1 | * | * | * | | + | + | |
| Groundwater abstractions (livestock) | *Groundwater abstractions* that are intended for livestock water use. | m3 m-2 month-1 | * | | * | | + | | |
| Groundwater demands | Gross water demand | m3 m-2 month-1 | * | * | * | | + | | |
| Groundwater depletion | Long-term losses from groundwater storage | m3 m-2 month-1 | * | * | | * | + | | * |
| Groundwater drainage/surface water capture | Exchange flux between groundwater and surface water. Groundwater leaving the aquifer = negative value; entering the aquifer = positive value | m3 m-2 month-1 | * | * | * | * | * | | * |
| Groundwater drainage/surface water capture from lakes | Exchange flux between groundwater and surface water (lakes); if available, additional to the sum of exchange fluxes (Groundwater drainage/surface water capture) also separate components can be provided/ Leaving the aquifer = negative values; entering the aquifer = positive value. | m3 m-2 month-1 | | | | * | * | | * |
| Groundwater drainage/surface water capture from rivers | Exchange flux between groundwater and surface water (rivers); if available, additional to the sum of exchange fluxes (Groundwater drainage/surface water capture) also separate components can be provided/ Leaving the aquifer = negative values; entering the aquifer = positive value. | m3 m-2 month-1 | | * | | * | * | | * |
| Groundwater drainage/surface water capture from springs | Exchange flux between groundwater and surface water (springs); if available, additional to the sum of exchange fluxes (Groundwater drainage/surface water capture) also separate components can be provided/ Leaving the aquifer = negative | m3 m-2 month-1 | | | | * | * | | * |

| | Description | Units | | | | | | | | |
|---|---|---|---|---|---|---|---|---|---|---|
| | values; entering the aquifer = positive value. | | | | | | | | | |
| Groundwater drainage/surface water capture from wetlands | Exchange flux between groundwater and surface water (wetlands); if available, additional to the sum of exchange fluxes (Groundwater drainage/surface water capture) also separate components can be provided/ Leaving the aquifer = negative values; entering the aquifer = positive value. | m3 m-2 month-1 | | | | * | * | | | * |
| Groundwater return flow | Return flow of abstracted groundwater (not yet separated into different sources). | m3 m-2 month-1 | * | * | | . | | . | | * |
| Groundwater storage | Mean monthly water storage in groundwater layer in kg m$^{-2}$. The spatial resolution is 0.5° grid. | m3 m-2 month-1 | * | * | | * | * | | | * |
| Hydraulic head | Head above sea level in m. If more than one aquifer layer is simulated, report the heads on the top productive aquifer (confined or unconfined). | m | * | | | * | * | | | * |
| Lateral groundwater flux (front face) | Cell-by-cell flow (front) | m3 m-2 month-1 | * | * | | * | * | | | * |
| Lateral groundwater flux (right face) | Cell-by-cell flow (right) | m3 m-2 month-1 | * | * | | * | * | | | * |
| Lateral groundwater flux (net) | Net cell-by-cell flow | m3 m-2 month-1 | * | * | | * | * | | | * |
| Lateral groundwater flux (lower face) | Cell-by-cell flow (lower) when more than 1 groundwater layer is simulated. | m3 m-2 month-1 | * | | | * | * | | | * |
| Submarine groundwater discharge | Flow of groundwater into oceans. The definition may vary by model. But in principle also models without density driven flow can submit this variable. | m3 m-2 month-1 | * | | | * | | | | * |
| Water table depth | Depth to the water table below land surface (digital elevation mode, DEM) in m. | m | * | * | | * | * | | | * |
| **Number of groundwater output** | Counting only currently available | | **19** | **13** | **9** | **14** | **14** | **1** | **1** | **17** |

| variables in model | | | | | | | | | | |
|---|---|---|---|---|---|---|---|---|---|---|
| | | 12 | | | | | | | | |



**4 Unstructured experiments point out model differences that should be explored further**

The ISIMIP groundwater sector is in an early development stage, and we hope that an ensemble of groundwater
models driven by the same meteorological data will be available soon. Yet, to provide first insights into the
models, their outputs, and how these can be compared, we collected existing outputs from the participating models
(see Table A1 for an overview). We opted for a straightforward initial comparison due to the various data formats,
model resolutions, and forcings that complicate a more thorough examination of a specific scientific inquiry. One
of our goals in the Groundwater sector is to conduct extensive analysis to better illustrate and understand the
model differences. The analysis presented here is intended solely as an introductory overview to provide a sense
of the rationale behind our initiative. Some overlap with recent model comparison studies naturally exists (e.g.,
Gnann et al., 2023; Reinecke et al., 2024, Reinecke et al. 2021); however, the presented analysis contains a
different ensemble of models and thus provides new insights. Hence, this descriptive analysis serves as an
introductory overview that highlights the present state of the art and identifies model discrepancies warranting
further investigation. In addition, relevant output data are not yet available for all models. We focused on the two
variables with the largest available ensemble: water table depth (G³M, CLM, WBM, and VIC-wur; Table 1) and
groundwater recharge (CLM, CWatM, GGR, VIC-wur, V2KARST, WBM; Table 1), only on historical periods
rather than future projections.
The arithmetic mean (not weighted by cell area) global water table depth varies substantially (6 m – 127 m)
between the models at the start of the simulation (1980 or steady-state) (Fig. 1a). On average, the water table of
G³M (28 m) and CLM (6 m) are shallower than WBM (127 m) and VIC-wur (81 m), whereas the latter two also
show a larger standard deviation (WBM: 133 m, VIC-wur: 105 m) than the other two models (G³M: 49 m, CLM:
3 m). The consistently shallower WTD of CLM impacts the ensemble mean WTD (Fig. 1b), which is shallower
compared to other model ensembles (5.67 m WTD as global mean here compared to 7.03 m in Reinecke et al.
248 (2024)).

This difference in ensemble WTD points to conceptual differences between the models. G³M and CLM both use
the relatively shallow WTD estimates of Fan et al. (2013) as initial state or spin-up, which could explain the
overall shallow water table depth. The difference between G³M and VIC-wur is consistent with the findings in
Reinecke et al. (2024), which showed a deeper water table simulated by the de Graaf et al. (2017) groundwater
model, which developed an aquifer parameterization adapted and conceptually similar to VIC-wur and WBM.
This difference may be linked to the implementation of groundwater drainage/surface water infiltration or
transmissivity parameterizations (Reinecke et al., 2024) as well as differences in groundwater recharge (Reinecke
et al., 2021). Furthermore, the models are not yet driven by the same climatic and human forcings, thereby possibly
causing different model responses. The newly initiated ISIMIP Groundwater sector offers an opportunity to
investigate these differences much more systematically in future studies, for example, by ruling out forcing as a
driver of the model differences and by exploring spatial and temporal relationships with key groundwater drivers
such as topography (e.g., Reinecke et al., 2024). In addition, the ISIMIP Groundwater sector provides a platform
for using the modelling team's expertise on their model implementations (e.g., model structures and parameter
fields) to better understand the origins of these differences.

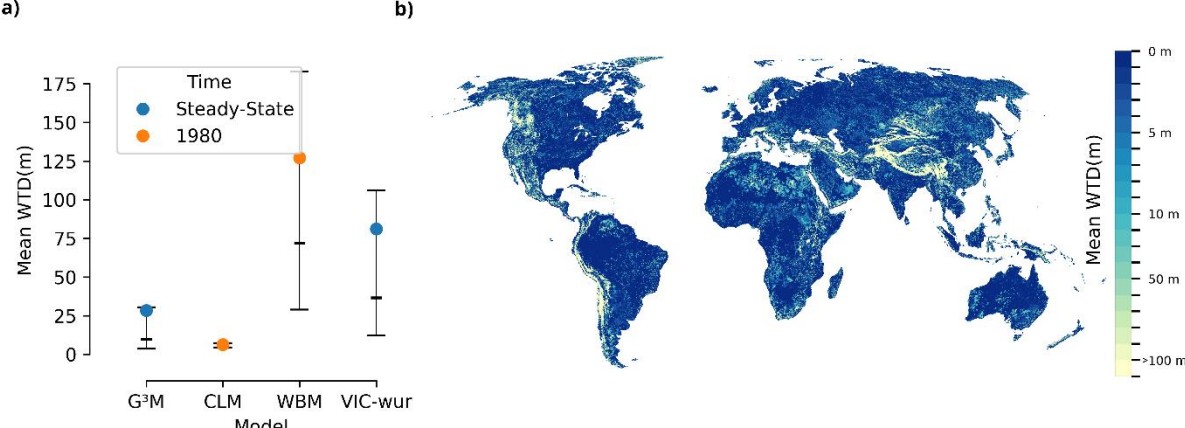

**Figure 1**: Global water table depth (WTD) at simulation start (1980) or the used steady-state. The simplified boxplot (a) shows the arithmetic model mean as a colored dot and the median as a black line. Whiskers indicate the 25[th] and 75[th] percentiles, respectively. The global map (b) shows the arithmetic mean of the model ensemble. Models shown are not yet driven by the same meteorological forcing (see also table A1).

Similarly, the global arithmetic mean groundwater recharge (not weighted by cell area) differs by 332 mm/y between models (150 mm/y excluding V2KARST since it calculates recharge in karst regions only) (Fig. 2a). This difference in recharge is more pronounced spatially (Fig. 2b) than differences in WTD shown before (Fig. 1b). Especially in drier regions such as in the southern Africa, central Australia, and the northern latitudes show coefficient of variation of 1 or greater (white areas). In extremely dry areas such as the east Sahara and southern Australia, the model spread is close to 0 (dark green). While the agreement is higher in Europe and western South America, the global map differs slightly from other recent publications (e.g., compared to Fig. 1b in Gnann et al. (2023)). In light of other publications, highlighting model uncertainty in groundwater recharge (Reinecke et al., 2021, Kumar et al., 2025) and the possible impacts of long-term aridity changes on groundwater recharge (Berghuijs et al., 2024), an extended combined ensemble of the global water sector and the new Groundwater sector could yield valuable insights.

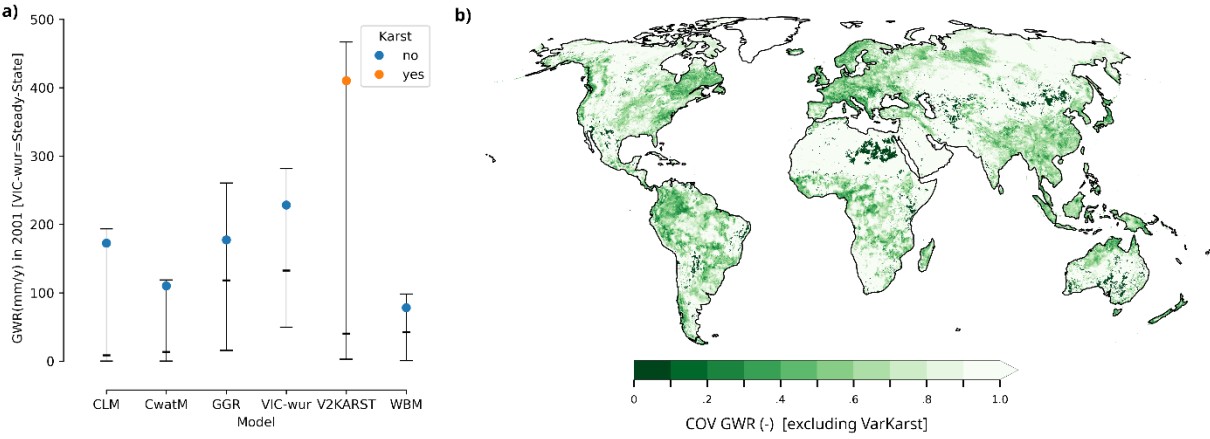

**Figure 2:** Global groundwater recharge (GWR) in 2001 or at steady-state (only VIC-wur). The simplified boxplot (a) shows the arithmetic model mean as a colored dot and the median as a black line. Whiskers indicate the 25[th]

and 75th percentiles, respectively. The global map (b) shows the coefficient of variation of the model ensemble
without V2KARST calculated as the ensemble standard deviation divided by the ensemble mean. Models shown
are not yet driven by the same meteorological forcing (see also table A1).

We further calculated relative changes in groundwater recharge between 2001 and 2006 (Fig. 3) with an ensemble
of 7 models (CLM, CWatM, GGR, VIC-wur, V2KARST, WBM, and ParFlow). The ensemble includes two
models that only simulate specific regions (V2KARST: regions of karstifiable rock, ParFlow: Euro CORDEX
domain). This result shows a potential analysis that should be repeated within the new Groundwater sector.
Intentionally, we do not investigate model agreement on the sign of change or compare them with observed data.
The ensemble still highlights plausible regions of groundwater recharge changes, such as in Spain and Portugal,
which aligns with droughts in the investigated period (Paneque Salgado and Vargas Molina, 2015; Coll et al.,
2017; Trullenque-Blanco et al., 2024). Relative increases in groundwater recharge are mainly shown for arid
regions in the Sahara, the Middle East, Australia, and Mexico. However, it is likely that because we investigate
relative changes, this might be related to the already low recharge rates in these regions.

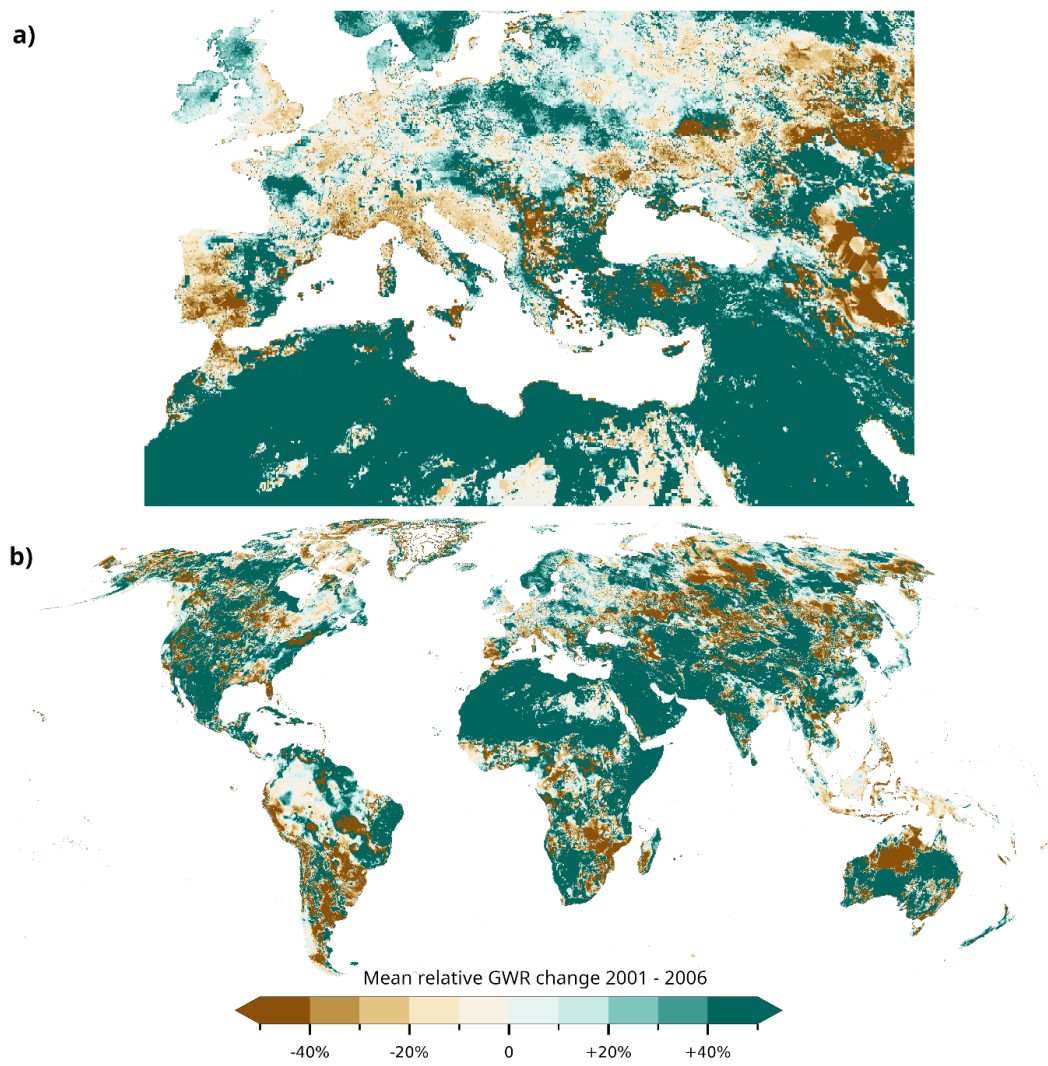


**Figure 3**: Mean relative percentage change of yearly groundwater recharge between 2001 and 2006 for Europe
(a), and all continents except Antarctica (b). The ensemble consists of all models that provided data for the years
2001 and 2006 (CLM, CWatM, GGR, VIC-wur, V2KARST, WBM, and ParFlow). V2KARST (only karst) and
ParFlow (only Euro CORDEX domain) were only accounted for in regions where data is available. Models shown
are not yet driven by the same meteorological forcing (see also table A1).

**5 Groundwater as a linking sector in ISIMIP**
ISIMIP encompasses a wide variety of sectors. Currently, 18 sectors are part of the impact assessment effort. The
Groundwater sector offers a new and unique opportunity to enhance cross-sectoral activities within ISIMIP, foster
interlinkages within ISIMIP, and thus deliver interdisciplinary assessments of climate change impacts.

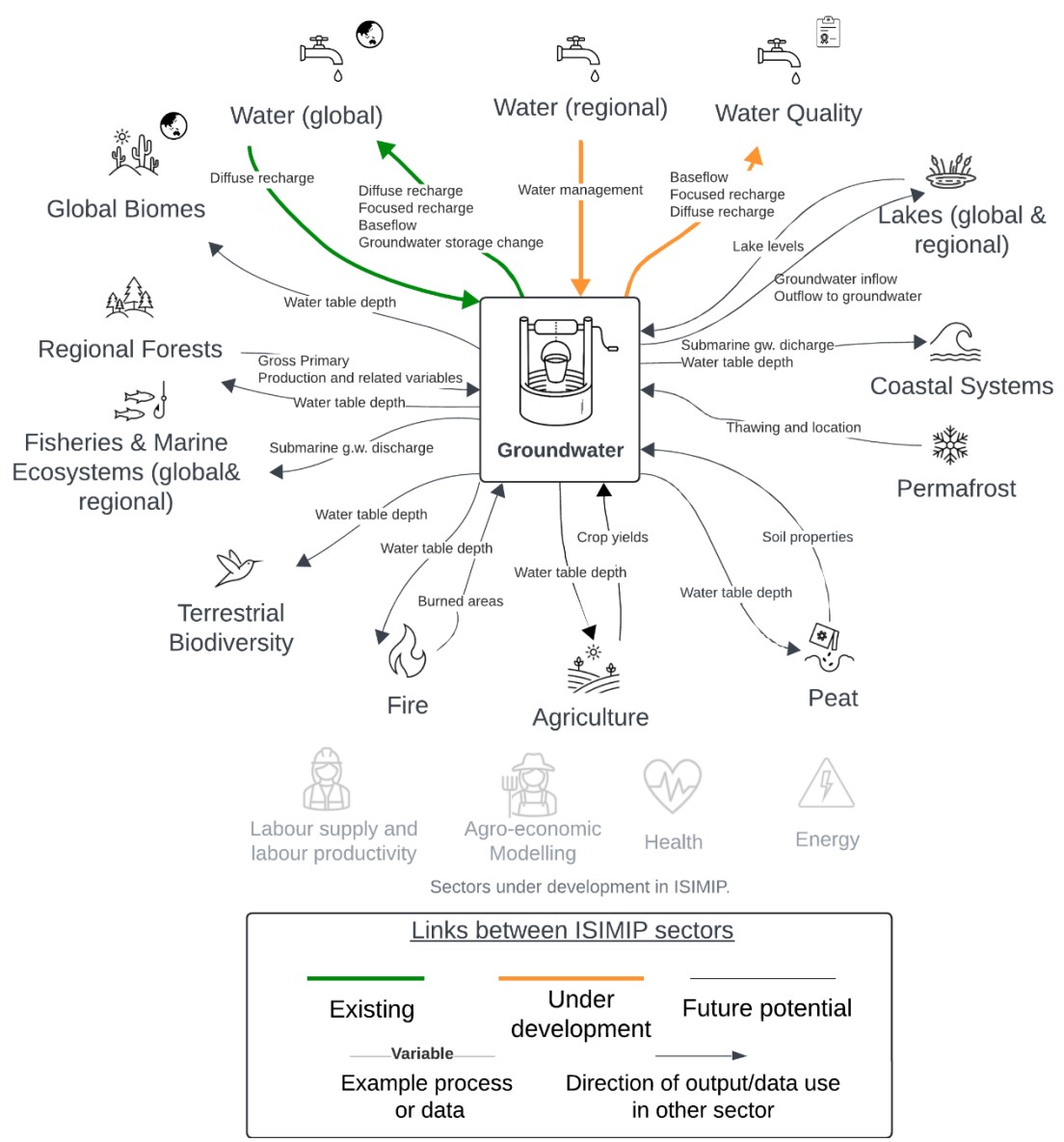


**Figure 4**: The Groundwater sector provides the potential for multiple interlinkages between different sectors
within ISIMIP. In the coming years, we will focus on links to three sectors (green and orange): Water (global),
Water (regional), and Water Quality. Other cross-sectoral linkages between non-Groundwater sectors (i.e.,
linkages between the outer circle) are not shown. Sectors that are currently under development or have not yet
have data or outputs that could be shared or used for cross-sectoral assessments are shown in gray. Interactions
between sectors are annotated with example processes, key variables, or datasets that can be shared between
sectors.
Some links with other sectors within ISIMIP are more evident than others with regard to existing scientific
community overlaps or existing scientific questions (Fig. 4). The examples of variables and data that can be shared
among sectors shown in Fig. 4 provide a non-exhaustive description of current variables that the sectors already
describe in their protocols. Whether cross-sectoral assessments will utilize this available data is up to the modeling
teams that contribute to the sectors. For example, the new Groundwater sector will focus on large-scale
groundwater models, some of which are already part of global water models participating in the Global Water
Sector or using outputs (such as groundwater recharge) from the Global Water Sector (see also existing
groundwater variables in the global water sector Table A2). However, the Groundwater sector will also feature
non-global representations of groundwater. Thus, collaborating with the Regional Water sector could provide
opportunities to share outputs and pursue common assessments. For example, the outputs of the groundwater
model ensemble, such as water table depth variations or surface water groundwater interactions, could be used as
input for some regional models that consider groundwater only as a lumped groundwater storage. Conversely,
global and continental groundwater models can benefit from validated regional hydrological models, which may
provide valuable insights into local runoff generation processes and the impacts of water management.
Furthermore, the relevance of groundwater for water quality assessments is widely recognized (e.g., for
phosphorous transport from groundwater to surface water (Holman et al., 2008), or for salinization (Kretschmer
et al., 2025), or as a link between warming groundwater and stream temperatures (Benz et al., 2024). And the
community effort of Friends of Groundwater called for a global assessment of groundwater quality (Misstear et
al., 2021). The Water Quality sector could incorporate model outputs from the Groundwater sector as input to
improve, for example, their estimates of groundwater contributions to surface water quantity or leakage of surface
water to groundwater. On the other hand, the Groundwater sector can utilize estimates of the Water Quality sector
to better assess water availability by incorporating water quality criteria. Ultimately, this may also result in
advanced groundwater models in the Groundwater sector that account for quality-related processes directly, which
can then be integrated into a future modeling protocol. One of the models (G³M; see Table 1) is already capable
of simulating salinization processes.
Leveraging such connections between sectors will provide valuable insights beyond groundwater itself. The
outputs and models that can be used for intersectoral assessments depend on the research question and may
necessitate the use of only a subset of models from an ensemble. Specifically, considering groundwater quality, a
collaboration between both sectors could be achieved in multiple aspects. Integrating groundwater availability
with water quality helps ensure sufficient and safe drinking and irrigation water. Focusing on aquifer storage
levels and pollutant loads can help maintain groundwater resilience, safeguard food security, and protect public
health under changing climate and socioeconomic conditions. Further, integrating groundwater quantity data with
pollution source mapping helps prioritize remediation efforts where aquifers are most vulnerable, ensuring both
water availability and quality. Concerning observational data, a unified approach to collecting and developing
shared databases for groundwater levels and water quality measurements across multiple agencies reduces
bureaucratic hurdles and ensures consistent, comparable data. Using standardized procedures for dealing with
observational uncertainties, such as data gaps, scaling issues, and measurement inconsistencies, would support
collaborative research further.
Research opportunities arise in other sectors as well. Groundwater is connected to the water cycle and social,
economic, and ecological systems (Huggins et al., 2023). For example, health impacts (such as water- and vector-
borne diseases) are closely related to water quantity and quality (e.g. Smith et al. (2024)), and the roles of
groundwater for forest resilience (regional forest sector, (Costa et al., 2023; Esteban et al., 2021)) and forest fires
(fire sector) under climate change are yet to be explored (Fig. 4). To prioritize our efforts and set a research agenda
for the groundwater ISIMIP sector, we will first focus on existing and more straightforward connections to the
global water sector, regional water sector, and the water quality sector and then expand to collaboration with other
sectors (Fig. 4).

**6 A vision for the ISIMIP groundwater sector**

Given groundwater's importance in the Earth system and for society, it is imperative to expand our knowledge of
groundwater and (1) how it is impacted by climate change and other human forcings and (2) how, in turn, this
will affect other systems connected to groundwater. This enhanced understanding is essential to equip us with the
knowledge needed to address future challenges effectively. The ISIMIP Groundwater sector serves as a foundation
for examining and measuring the effects of global change on groundwater systems worldwide. It facilitates cross-
sector investigations, such as those concerning water quality, examines the influence of various model structures
on groundwater dynamics simulations, and supports the collaborative creation of new datasets for model
parameterization and assessment. Other intercomparison and impact assessment projects already have been
successful in achieving similar goals such as the lake (Golub et al., 2022) or water quality sector (Strokal et al.,
2025) in ISIMIP, the CMIP (Eyring et al., 2016a), or the AgMIP for agricultural models (von Lampe et al., 2014).
Already in the short term, the creation of the Groundwater sector has substantial potential to enhance large-scale
groundwater research by developing better modeling frameworks for reproducible research (running the multitude
of experiments targeted in ISIMIP requires an automated modeling pipeline) and forge a community that can
critically examine current modeling practices. The simple model comparison presented raises initial questions as
to why models differ and invites us to explore model differences in greater depth. Such model intercomparison
studies will enable us to quantify uncertainties and identify hotspots for model improvement. They will also allow
us to assess the impact of climate and land use change on various groundwater-related variables, such as
groundwater recharge and water table depth, and enable ensemble-based impact assessments of future water
availability. Model intercomparison and validation may also help identify models that perform better in specific
regions or for specific output variables, thus allowing the provision of region- or variable-specific
recommendations and uncertainty assessments to subsequent data users.
In the long term, the sector will enable us to jointly reflect on processes that we currently do not model or that
require improvement, possibly also through new modeling approaches such as hybrid machine-learning models
tailored to the large-scale representation of groundwater. These model developments will be incorporated into the
groundwater sector's contributions to upcoming ISIMIP simulation rounds, such as ISIMIP4, which is scheduled
to commence in 2026. Since groundwater is connected to many socio-ecological systems, groundwater models
could also emerge as a modular coupling tool that can be integrated into multiple sectors. The newly established
groundwater sector already provides a first step in that direction by standardizing output names and units. If
models are modular enough and define a standardized Application Programming Interface (API), they could also
serve as a valuable tool for other science communities.
The lack of a community-wide coordinated effort to simulate the effects of climate change on groundwater at
regional to global scale has precluded the comprehensive consideration of climate change impacts on groundwater
in policy relevant reports, such as the European Climate risk assessment (EUCRA, 2024) or the Assessment
Reports developed by the Intergovernmental Panel on Climate Change (IPCC) (e.g. Lee, 2024). The anticipated
groundwater sector contributions to ISIMIP3 and ISIMIP4, as described here, will address this gap by serving as
scientific evidence in the second EUCRA round and the upcoming IPCC seventh assessment cycle. As such, the
anticipated outcomes of the new sector will pave the way for groundwater simulations to play an increasingly
important role in international climate mitigation and adaptation policy.
In summary, the ISIMIP Groundwater sector aims to enhance our understanding of the impacts of climate change
and direct human impacts on groundwater and a range of related sectors. To realize this goal, the new ISIMIP
Groundwater sector will address numerous challenges. For instance, core simulated variables, such as water table
depth and recharge, are highly uncertain and difficult to compare with observations. Further, tracing down
explanations for inter-model differences will require the joint development and application of new evaluation
methods (Eyring et al., 2016b) and protocols. Currently, models of the Groundwater sector operate at different
spatial resolutions, and compared to other sectors, they often run at relatively high spatial resolutions, which will
need to be addressed in evaluation and analysis approaches. Furthermore, depending on the model, executing
single-model simulations already requires substantial amounts of computation time, and running all impact
scenarios may be infeasible for some modeling groups. Lastly, running simulations for ISIMIP requires not only
computational resources but also human resources, which might not be feasible for all groups. This has always
been the case with ISIMIP, and it is an issue that other sectors have faced as well. Still, we are confident that the
groundwater sector will enhance our understanding of groundwater within the Earth system and help to promote
dialogue and synthesis in the research community. With its various connections to other sectors, the Groundwater
sector can be a catalyst for developing new holistic cross-sector modelling efforts that account for the multitude
of interconnections between the water cycle and social, economic, and ecological systems.
**Data availability**
The ensemble-mean WTD and groundwater recharge trends are available in Reinecke (2025b)
https://doi.org/10.5281/zenodo.14962511. The Zenodo repository included pre-processing scripts, plotting
files, and data, as well as the main outputs presented in this manuscript as raster files. For the original model data
publications, see Table A1.

**Author contribution**
RR led the writing and analysis of the manuscript. RR and IG conceived the idea. All authors reviewed the
manuscript and provided suggestions on text and figures.

**Competing interests**
None.

**Appendix**
**Table A1**: Original publications that describe the model outputs used in section 4.

| Model | Simulation setup and used forcings | Reference |
|---|---|---|
| G³M | Steady-state model of WTD on 5 arcmin without any groundwater pumping, forced with WaterGAP 2.2d (Müller Schmied et al., 2021) groundwater recharge mean between 1901-2001. | Reinecke et al. (2019) |
| V2KARST | Global karst recharge model at 15 arcmin, forced with the MSWEP V2 (Beck et al., 2019) precipitation and GLDAS (Li et al., 2018) air temperature, shortwave and longwave radiation, specific humidity and wind speed for the period of 1990-2020 | Sarrazin et al. (2018) |
| GGR | Global groundwater rain-fed recharge model, A grid-based three-layer water balance model to estimate the daily global rain-fed groundwater recharge (2001-2020) | Nazari et al. (2025) |
| WBM | Time series simulation from 1980 to 2019 at 15 arc minutes, using the MERIT digital flow direction dataset (Yamazaki et al., 2019) including domestic, industrial, livestock, and irrigation water withdrawals. Forcings and key inputs: Climate: ERA5 (Prusevich et al., 2024), Reservoirs: GRanD v1.1 (Lehner et al., 2011), Inter-basin transfers (Lammers, 2022), Glaciers (Rounce et al., 2022), Impervious surfaces (Hansen and Toftemann Thomsen, 2020), Population density (Lloyd et al., 2019), Domestic and industrial water per capita demand: FAO | Multiple, see left column. |

| | AQUASTAT, Livestock density and water demand (Gilbert et al., 2018), Cropland: LUH2 (Hurtt et al., 2020), Aquifer properties (de Graaf et al., 2017) aquifer depth gap-filled with terrain slope data from Yamazaki et al. 2019, Soil available water capacity: FAO soil map, Root depth (Yang et al., 2016) | |
|---|---|---|
| VIC-wur | Global Hydrological model simulating the GWR and streamflow from 1970-2014 in natural condition.<br><br>The mean GWR and streamflow were used to simulate the GWT in steady-state MODFLOW model in 5 arcmin.<br><br>The model is forced by: GFDL-ESM4 climate model (Dunne et al., 2020), Aquifer properties (de Graaf et al., 2017). | Droppers et al. (2020) |
| CLM | The model was spun up for 1979 and subsequently simulated from 1979 to 2013 using the GSWPv3 atmospheric forcing dataset at a 0.1-degree resolution. Recharge, capillary rise, drainage, irrigation pumping and cell-to-cell lateral flow were simulated within the model. | Akhter et al. (2025) |
| ParFlow | The data provided here are based on Naz et al. (2023). In version 2 of the data, we provide variables including water table depth and groundwater recharge for time period of 1997-2006 at monthly time scale. | Naz et al. (2023) |
| CWatM | Community Water Model at 5 arcmin. Climate forcing with chelsa-W5E5v1.0 (5 arcmin) for temperature (average, maximum, minimum), precipitation, and shortwave radiation, and GSWP3-W5E5 (30 arcmin spline downscaled to 5 arcmin) for longwave radiation, wind speed, and specific humidity. Updates to Burek et al. (2020) include river network based on MERIT Hydro and upscaling with Eilander et al. (2021). | Burek et al. (2020) |


**Table A2**: List of groundwater related output variables in the ISIMIP3a global water sector (https://protocol.isimip.org/#/ISIMIP3a/water_global). The unit of all variables is kg m$^{-2}$ s$^{-1}$, the spatial resolution is 0.5° grid and the temporal resolution is monthly.

| Groundwater-related output variable of the Global Water Sector | Description |
|---|---|
| Groundwater runoff | Water that leaves the groundwater layer. In case seepage is simulated but no groundwater layer is present, report seepage as *Total groundwater recharge* and *Groundwater Runoff.* |
| Total groundwater recharge | For models that consider both diffuse and focused/localised recharge this should be the sum of both; other models should submit the groundwater recharge component that the model simulates. See also the descriptions in *Focused/localised groundwater recharge* and *Diffuse groundwater recharge.* |
| Focused/localised groundwater recharge | Water that directly flows from a surface water body into the groundwater layer below. Only submit if the model separates focused/localised recharge from diffuse recharge. |
| Potential irrigation water withdrawal (assuming unlimited water supply) from groundwater resources | Part of *Potential Industrial Water Withdrawal* that is extracted from groundwater resources. |
| Actual irrigation water withdrawal from groundwater resources | Part of *Actual Irrigation Water Withdrawal* that is extracted from groundwater resources. |
| Potential Irrigation Water Consumption from groundwater resources | Part of *Potential Irrigation Water Consumption* that is extracted from groundwater resources. |
| Actual Irrigation Water Consumption from groundwater resources | Part of *Actual Irrigation Water Consumption* that is extracted from groundwater resources. |
| Potential Domestic Water Withdrawal from groundwater resources | Part of *Potential Domestic Water Withdrawal* that is extracted from groundwater resources. |
| Actual Domestic Water Withdrawal from groundwater resources | Part of *Actual Domestic Water Withdrawal* that is extracted from groundwater resources |
| Potential Domestic Water Consumption from groundwater resources | Part of *Potential Domestic Water Consumption* that is extracted from groundwater resources. |

| | |
|---|---|
| Actual Domestic Water Consumption from groundwater resources | Part of *Actual Domestic Water Consumption* that is extracted from groundwater resources. |
| Potential Manufacturing Water Withdrawal from groundwater resources | Part of *Potential Manufacturing Water Withdrawal* that is extracted from groundwater resources. |
| Actual Manufacturing Water Withdrawal from groundwater resources | Part of *Actual Manufacturing Water Withdrawal* that is extracted from groundwater resources. |
| Potential manufacturing Water Consumption from groundwater resources | Part of *Potential manufacturing Water Consumption* that is extracted from groundwater resources. |
| Actual Manufacturing Water Consumption from groundwater resources | Part of *Actual Manufacturing Water Consumption* that is extracted from groundwater resources. |
| Potential electricity Water Withdrawal from groundwater resources | Part of *Potential electricity Water Withdrawal* that is extracted from groundwater resources. |
| Actual Electricity Water Withdrawal from groundwater resources | Part of *Actual Electricity Water Withdrawal* that is extracted from groundwater resources. |
| Potential electricity Water Consumption from groundwater resources | Part of *Potential electricity Water Consumption* that is extracted from groundwater resources. |
| Actual Electricity Water Consumption from groundwater resources | Part of *Actual Electricity Water Consumption* that is extracted from groundwater resources. |
| Potential Industrial Water Withdrawal from groundwater resources | Part of *Potential Industrial Water Withdrawal* that is extracted from groundwater resources. |
| Actual Industrial Water Withdrawal from groundwater resources | Part of *Actual Industrial Water Withdrawal* that is extracted from groundwater resources. |

| | |
|---|---|
| Potential Industrial Water Consumption from groundwater resources | Part of *Potential Industrial Water Consumption* that is extracted from groundwater resources. |
| Actual Industrial Water Consumption from groundwater resources | Part of *Actual Industrial Water Consumption* that is extracted from groundwater resources. |
| Potential livestock Water Withdrawal from groundwater resources | Part of *Potential livestock Water Withdrawal* that is extracted from groundwater resources. |
| Actual Livestock Water Withdrawal from groundwater resources | Part of *Actual Livestock Water Withdrawal* that is extracted from groundwater resources. |
| Potential livestock Water Consumption from groundwater resources | Part of *Potential livestock Water Consumption* that is extracted from groundwater resources. |
| Actual livestock Water Consumption from groundwater resources | Part of *Actual livestock Water Consumption* that is extracted from groundwater resources. |
| Total Potential Water Withdrawal (all sectors) from groundwater resources | Part of *Total Potential Water Withdrawal* that is extracted from groundwater resources. |
| Total Actual Water Withdrawal (all sectors) from groundwater resources | Part of *Total Actual Water Withdrawal* that is extracted from groundwater resources. |

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
