# Peer review of "The ISIMIP Groundwater Sector: A Framework for Ensemble"

_EGUsphere, 2025_

## Author Comment (AC1)

Dear Thomas Wild,

Thank you very much for handling our manuscript. Below, we respond to the two reviewers' feedback. As a result, we have improved the manuscript in multiple ways:

- We clarified the motivation of our sector and the community aspect of the creation of the sector.
- We expanded on possible challenges in the conclusions.
- We expanded the discussion of interlinkages to other sectors and improved Figure 4.
- And we improved on the discussion of the results and conclusions we drew.

In the spirit of a community project, we also added two additional authors: Wim Thiery and Tanjila Akhter, because they contributed to the conceptualization of the sector in previous workshops and this revision. Moreover, they are committed to providing model outputs to the ISIMIP Groundwater sector, utilizing the CLM model in their respective groups.

Below, we provide our responses to the comments in blue. To ensure a more streamlined integration of feedback from both reviewers, we chose to address reviewer 2's feedback first, followed by reviewer 1's.

On behalf of all authors,

Robert Reinecke

**Reviewer 2**

This paper provides an overview of the recently established groundwater sector within the Inter Sectoral Impact Modeling Intercomparison Project (ISIMIP). The paper is quite succinct, and for the most part, strikes a good balance of keeping the presentation high level while still being informative. A small amount of analysis is provided in Section 4 to offer an initial glimpse at notable differences in model outputs for water table depth and recharge, but the aim of this paper is to introduce the motivation for groundwater ISIMIP sector, some background on the participating models, and the sector's short to medium term vision. Overall, I think this paper provides a very approachable and well-presented overview of the new groundwater sector. I have some minor comments that could improve the manuscript, but no major criticisms that need to be addressed.

We thank the reviewer for taking the time to read our manuscript and appreciate the constructive feedback.

Minor comments:

- Section 6 could be improved by adding a few sentences about some potential challenges, technical, logistical, monetary (funding), for realizing the goals of new Groundwater MIP sector.

We have expanded section 6 with the following (line 384):

*"In summary, the ISIMIP Groundwater sector aims to enhance our understanding of the impacts of climate change and direct human impacts on groundwater and a range of related sectors. To realize this goal, the new ISIMIP Groundwater sector will address numerous challenges. For instance, core simulated variables, such as water table depth and recharge, are highly uncertain and difficult to compare with observations. Further, tracing down explanations for inter-model differences will require the joint development and application of new evaluation methods (Eyring et al., 2016b) and protocols. Currently, models of the Groundwater sector operate at different spatial resolutions, and compared to other sectors, they often run at relatively high spatial resolutions, which will need to be addressed in evaluation and analysis approaches. Furthermore, depending on the model, executing single-model simulations already requires substantial amounts of computation time, and running all impact scenarios may be infeasible for some modeling groups. Lastly, running simulations for ISIMIP requires not only computational resources but also human resources, which might not be feasible for all groups. This has always been the case with ISIMIP, and it is an issue that other sectors have faced as well. Still, we are confident that the groundwater sector will enhance our understanding of groundwater within the Earth system and help to promote dialogue and synthesis in the research community. With its various connections to other sectors, the Groundwater sector can be a catalyst for developing new holistic cross-sector modelling efforts that account for the multitude of interconnections between the water cycle and social, economic, and ecological systems."*

- Something not touched on is that certain models could perform better in specific regions on for specific output variables. There could be value in more directly stating that the Groundwater ISIMIP sector could inform region-specific model recommendations for specific outputs.

Thank you for this suggestion. We have added this aspect to section 6.

Line 364 now reads:

*"Model intercomparison and validation may also help identify models that perform better in specific regions or for specific output variables, thus enabling the provision of region- or variable-specific recommendations and uncertainty assessments to subsequent data users."*

- L 135: could the authors expand slightly on "functional relationships"?

We agree that the current explanation may be too brief. We have expanded on this as requested.

Line 155 now reads:

*"We aim to utilize these simulations for an in-depth model comparison, including a comparison to observational data such as time series of groundwater table depth (e.g., Jasechko et al. (2024)) and by utilizing so-called functional relationships (Reinecke et al., 2024; Gnann et al. 2023). Functional relationships can be defined as covariations of variables across space and/or time, and they are a key aspect of our theoretical knowledge of Earth's functioning. Examples include relationships between precipitation and groundwater recharge (Gnann et al. 2023; Berghuijs et al. 2024) or between topographic slope and water table depth (Reinecke et al., 2024)."*

- L 136-138 Groundwater is new to ISIMIP, but are there any notable previous groundwater model intercomparison efforts worth mentioning, even if not global in scale?

We are not aware of an intercomparison project that utilizes multiple groundwater models, especially for large-scale analyses. There are examples of benchmarking experiments that use different groundwater modeling approaches for the same problem or run the same model for different scenarios, but we are not aware of an effort, even on regional scales, that has approached such an intercomparison. We have added this discussion to the introduction and also added a reference to GroMoPo as requested by Reviewer 1.

Line 106 in the introduction now reads:

"[...] *The new Groundwater sector is a separate but complementary sector to the existing global water sector. To our knowledge, there are currently no long-term community efforts for a structured model intercomparison project for groundwater models. While studies have benchmarked different model approaches (e.g., Maxwell et al. 2014), or compared model outputs (Reinecke et al., 2021; 2024), or collected information on where and how we model groundwater (Telteu et al., 2021; Zipper et al., 2023; Zamrsky et al., 2025), no effort yet aims at forcing different groundwater models with the same climate and human forcings for different scenarios.*"

- L179-189: In addition to using the same forcing data, are there plans to run these models at the same resolution or are outputs going to be scaled to the same resolution as a post-processing step?

In the short term, there are no plans to run the models at the same resolutions, as this might require substantial changes in some models or may not even be possible for others because of computation time and resource limitations. Model outputs are scaled to the same resolution as a post-processing step, which makes it more feasible to conduct model intercomparisons. In this way, more groundwater models are likely to participate. However, some of the models already run on the same resolution or are flexible in running on multiple spatial and temporal resolutions – it would be worth investigating the differences in scale sensitivity.

As the sector further develops in the future, it may become possible and desirable to harmonize spatial resolution for improved inter-comparability of model outputs.

We now clarify this in section 3 (line 190):

"The current sector protocol defines a targeted spatial resolution of 5 arcmin, as this represents not only the resolution achievable by most global models but also the coarsest resolution at which meaningful representation of groundwater dynamics, particularly lateral groundwater flows and water table depths, can still be captured (Gleeson et al., 2021). ISIMIP3 also specifies experiments with different spatial resolutions, but whether this is achievable with a sub-ensemble of the presented models remains unclear, as it depends on the available computational time, flexibility of model setups, and data availability. To ensure consistency and comparability, the model outputs are currently post-processed by the modeling groups to aggregate their outputs to the protocol-specified spatial and temporal resolutions."

- Figure 1 & Lines 195-196: "This difference in ensemble WTD points to conceptual differences between the models, which should be investigated further." Given the authors on this paper have expertise with these models, could there be a couple sentences offered positing as to why there could be such stark differences in the water depth for G3M & CLM compared to WBM and VIC-wur?

We agree that a more in-depth discussion of these differences is necessary, which is a key motivation for future research. However, it is also challenging to pinpoint precisely why the presented models differ so much without a much more thorough analysis. To provide better guidance for subsequent studies, we tried to expand on this aspect as much as possible. Line 234 now read:

*"This difference in ensemble WTD points to conceptual differences between the models. $G^3M$ and CLM both use the relatively shallow WTD estimates of Fan et al. (2013) as initial state or spin-up, which could explain the overall shallow water table depth. The difference between $G^3M$ and VIC-wur is consistent with the findings in Reinecke et al. (2024), which showed a deeper water table simulated by the de Graaf et al. (2017) groundwater model, which developed an aquifer parameterization adapted and conceptually similar to VIC-wur and WBM. This difference may be linked to the implementation of groundwater drainage/surface water infiltration or transmissivity parameterizations (Reinecke et al., 2024) as well as differences in groundwater recharge (Reinecke et al., 2021). Furthermore, the models are not yet driven by the same climatic and human forcings, thereby possibly causing different model responses. The newly initiated ISIMIP Groundwater sector offers an opportunity to investigate these differences much more systematically in future studies, for example, by ruling out forcing as a driver of the model differences and by exploring spatial and temporal relationships with key groundwater drivers such as topography (e.g., Reinecke et al., 2024). In addition, the ISIMIP Groundwater sector provides a platform for using the modelling team's expertise on their model implementations (e.g., model structures and parameter fields) to better understand the origins of these differences."*

- Figures 2 and 3: It is my understanding that for this initial comparison the groundwater recharge model results have different forcings (Table A1). I think it would be good to remind the audience of this because the rest of the paper is focused on the forthcoming efforts using the *same* forcings for the MIP.

Thank you for this suggestion. We have added the following sentence to all figures: "Models shown are not yet driven by the same meteorological forcing (see also table A1)."

Minor spelling comments:

- Spelling L150: "and surface water exchange fluxes as upper boundary conditionals without later fluxes": Later**al** missing "al"

Thank you for spotting this, we corrected it.

- Line 275: ISMIP missing I, ISIMIP

Thank you for spotting this, we corrected it.

**Reviewer 1**

I appreciate the efforts of Reinecke et al, and support the effort to better represent groundwater in ISIMIP - this is a much needed, and long called for effort. But overall this manuscript feels a thin, uncritical, non-exhaustive and somewhat repetitive. This may strongly worded, but it feels more like a paper written quickly after a great workshop rather than a deep effort with longer rumination and iteration. The manuscript seems thin in that each section seems quick and brief rather than deeply insightful or critical. I think a number of the ideas could be expanded upon with more critique and reflection. For example, when I look at the models in Table 2 compared to the linkages in Figure 4, I was struck by the limited capacity of most models to simulate outputs that would be useful for other sectors. At a basic level, if water use is not even in a model, how is it useful to assess water resources? And nothing to do with groundwater quality or contamination is mentioned in Table 1 so how can this effort be useful for water quality?

Deciding on a multi-model intercomparison protocol is not a trivial task. We thank the reviewer for the thoughtful feedback and for taking the time to read our manuscript. However, we would also like to point out that the assessment of the paper quality stands in contrast to the evaluation of Reviewer 2: "*This paper provides an overview of the recently established groundwater sector within the Inter Sectoral Impact Modeling Intercomparison Project (ISIMIP). The paper is quite succinct, and for the most part, strikes a good balance of keeping the presentation high level while still being informative.*"

We appreciate that the reviewer recognizes the success of our workshops; however, we would like to clarify that the paper is the outcome of multiple in-person workshops, held in Potsdam (2022, 2025), Prague (2023), Mainz (2024), as well as a EGU splinter meeting (2024) and numerous uncounted online meetings.

We believe that some of the comments stem from a lack of explanation for the motivation behind ISIMIP and the newly created sector. The critical comments thus helped to improve the manuscript. To improve our manuscript, we have in particular clarified the motivation behind our sector and the community aspect of its creation, expanded the discussion of interlinkages to other sectors, improved Figure 4, and expanded on the discussion of model differences.

To specifically address "if water use is not even in a model, how is it useful to assess water resources? And nothing to do with groundwater quality or contamination is mentioned in Table 1 so how can this effort be useful for water quality?"

Water uses are included in some models but not in all models. Furthermore, to better understand water resources, it is essential to comprehend the processes and interactions within the water cycle. This better understanding does not necessarily involve only the 'scenario' of water uses. ISIMIP also defines scenarios of "naturalized" conditions without human intervention. Some models can simulate multiple scenarios, while others may not.

Water quality is omitted here because global groundwater quality data remain sparse. Even surface water quality studies at this scale are nascent, prompting initiatives like ISIMIP's new water quality sector (see https://www.isimip.org/about/#sectors-and-contacts) to address this gap (with currently a paper on its protocol in the second round of review with ERL-Water).

We, however, acknowledge that scientists less familiar with or not involved in the ISIMIP setup may need further explanation on how community decision-making informs the creation of a sector and potential connections to other sectors. We have also expanded the description of ISIMIP to clarify the independence of sectors and models.

Line 139 now reads:

*"The creation of a new ISIMIP Groundwater sector is not linked to any funding and is a community-driven effort that includes all modeling groups that wish to participate. During the creation process, multiple groups and institutions were contacted to participate, and additional modeling groups are welcome to join the sector in the future. Models participating in the sectors do not need to be able to model all variables and scenarios defined in the protocol. ISIMIP sectors can be linked to broader thematic concepts, such as Agriculture, or can focus on specific components of the Earth system, such as Lakes or Groundwater. We would like to note that groundwater is not an isolated system, but rather part of the water cycle and the Earth system as a whole. Focusing on it within a dedicated sector aligns well with the existing models and is useful for studying groundwater systems in a thematically focused way. Collaboration (and perhaps integration) with sectors like the Global Water sector is possible and desirable in the future. We discuss possible future synergies with other existing ISIMIP sectors in Section 5."*

We also clarified that the usage of outputs for other sectors depends on the scientific question and specified how the quality sector might use outputs from the Groundwater sector in the future in section 5 (line 314):

*"Furthermore, the relevance of groundwater for water quality assessments is widely recognized (e.g., for phosphorous transport from groundwater to surface water (Holman et al., 2008), or for salinization (Kretschmer et al., 2025), or as a link between warming groundwater and stream temperatures (Benz et al., 2024). And the community effort of Friends of Groundwater called for a global assessment of groundwater quality (Misstear et al., 2021). The Water Quality sector could incorporate model outputs from the Groundwater sector as input to improve, for example, their estimates of groundwater contributions to surface water quantity or leakage of surface water to groundwater. On the other hand, the Groundwater sector can utilize estimates of the Water Quality sector to better assess water availability by incorporating water quality criteria. Ultimately, this may also result in advanced groundwater models in the Groundwater sector that account for quality-related processes directly, which can then be integrated into a future modeling protocol. One of the models ($G^3M$; see Table 1) is already capable of simulating salinization processes.*

*Leveraging such connections between sectors will provide valuable insights beyond groundwater itself. The outputs and models that can be used for intersectoral assessments depend on the research question and may necessitate the use of only a subset of models from an ensemble. Specifically, considering groundwater quality, a collaboration between both sectors could be achieved in multiple aspects. Integrating groundwater availability with water quality helps ensure sufficient and safe drinking and irrigation water. Focusing on aquifer storage levels and pollutant loads can help maintain groundwater resilience, safeguard food security, and protect public health under changing climate and socioeconomic conditions. Further, integrating groundwater quantity data with pollution source mapping helps prioritize remediation efforts where aquifers are most vulnerable, ensuring both water availability and quality. Concerning observational data, a unified approach to collecting and developing shared databases for groundwater levels and water quality measurements across multiple agencies reduces bureaucratic hurdles and ensures consistent, comparable data. Using standardized procedures for dealing with observational uncertainties, such as data gaps, scaling issues, and measurement inconsistencies, would support collaborative research further."*

Section 4 about unstructured experiments seemed repetitive to other recent articles on uncertainty in the water table depth and recharge including those of co-authors. It also felt thin and preliminary, and frankly uninspiring (in that the models seem to show little consistency) and unsurprising (due to overlap with previous articles).

We thank the reviewer for this comment. It is true that this section provides only limited new insights, but this was not the primary aim here, as stated in the original text: *"We opted for a straightforward initial comparison due to the various data formats, model resolutions, and forcings that complicate a more thorough examination of a specific scientific inquiry. Thus, this descriptive analysis serves as an introductory overview that highlights the present state of the art and identifies model discrepancies warranting further investigation."*

To clarify this further, we added "One of our goals in the Groundwater sector is to conduct extensive analysis to better illustrate and understand the model differences. The analysis presented here is intended solely as an introductory overview to provide a sense of the rationale behind our initiative."

In response to Reviewer 2, we also improved the discussion of the model differences.

Line 234 now read:

*"This difference in ensemble WTD points to conceptual differences between the models. $G^3M$ and CLM both use the relatively shallow WTD estimates of Fan et al. (2013) as initial state or spin-up, which could explain the overall shallow water table depth. The difference between $G^3M$ and VIC-wur is consistent with the findings in Reinecke et al. (2024), which showed a deeper water table simulated by the de Graaf et al. (2017) groundwater model, which developed an aquifer parameterization adapted and conceptually similar to VIC-wur and WBM. This difference may be linked to the implementation of groundwater drainage/surface water infiltration or transmissivity parameterizations (Reinecke et al., 2024) as well as differences in groundwater recharge (Reinecke et al., 2021). The newly initiated ISIMIP Groundwater sector offers an opportunity to investigate these differences much more systematically in future studies, for example, by ruling out forcing as a driver of the model differences and by exploring spatial and temporal relationships with key groundwater drivers such as topography (e.g., Reinecke et al., 2024). In addition, the ISIMIP Groundwater sector provides a platform for using the modelling team's expertise on their model implementations (e.g., model structures and parameter fields) to better understand the origins of these differences."*

In addition, we would like to highlight two novel aspects that differ from the previously published results of the co-authors. Here, we present a different set of models involved compared to Reinecke et al. (2024) (specifically WBM and CLM, and in part also VIC-Wur even if it is conceptually similar to the de Graaf model discussed in Reinecke et al. 2024), and we include models that specifically incorporate processes not included in previous assessments by Reinecke et al., such as karst. We thus also added the following statement in line 219:

*"Some overlap with recent model comparison studies naturally exists (e.g., Gnann et al., 2023; Reinecke et al., 2024, Reinecke et al. 2021), even though this brief analysis contains a different ensemble of models and thus provides new insights."*

Examples of it not being exhaustive is that it does not even mention the recent GroMoPo effort that a number of the authors have been involved with (Zipper et al. 2023; Zamrsky et al., 2025).

This initiative has compiled hundreds of regional scale model even though line 98 claims to 'integrate currently available groundwater models that operate at regional scale'.

We thank the reviewer for the suggestion to cite the GroMoPo project. Importantly, while GroMoPo collected information about groundwater models, it did not collect the models themselves or any knowledge about how to operate them. Thus, they do not provide any basis for being used in a model ensemble. The ISIMIP sector, of course, is open to any modeling groups that would still like to join the intercomparison initiative. We agree, however, that the phrasing in line 98 can be misleading, as the current sector integrates models of modeling teams that decided to commit their time to joint experiments.

We now cite GroMoPo in the introduction and specify why it is not directly helpful for the creation of this sector. Line 107 now reads:

*"To our knowledge, there are currently no long-term community efforts for a structured model intercomparison project for groundwater models. While studies have benchmarked different modeling approaches (e.g., Maxwell et al. 2014), compared model outputs (Reinecke et al., 2021; 2024), or collected information on where and how we model groundwater (Telteu et al., 2021; Zipper et al., 2023; Zamrsky et al., 2025), no effort yet aims at forcing different groundwater models with the same climate and human forcings for different scenarios."*

And we have rephrased the initial sentence of the paragraph to be more specific:

Line 100 now reads:

*"Here, we present a new sector in ISIMIP called the ISIMIP Groundwater Sector, which integrates models of the groundwater community that operate at regional (at least multiple km² (Gleeson and Paszkowski, 2014)) to global scales and are committed to providing model simulations to this new sector."*

Also missing are any mention of linking with global groundwater quality and contamination efforts such as Friends of Groundwater which seems important for the groundwater quality linkage.

Thank you for this suggestion. We are now citing the Friends of Groundwater initiative in line 317:

*"Furthermore, the relevance of groundwater for water quality assessments is widely recognized (e.g., for phosphorous transport from groundwater to surface water (Holman et al., 2008), or for salinization (Kretschmer et al., 2025), or as a link between warming groundwater and stream temperatures (Benz et al., 2024). And the community effort of Friends of Groundwater called for a global assessment of groundwater quality (Misstear et al., 2021). The Water Quality sector could incorporate model outputs from the Groundwater sector as input to improve, for example, their estimates of groundwater contributions to surface water quantity or leakage of surface water to groundwater."*

Finally, I was a recent reviewer of this manuscript by Huggins et al. (again with some of the same coauthors) and am struck that many of the linkages to other sectors would be much better created by taking a more holistic, social-ecological systems approach or at least bringing in insights and data from this approach than the narrow hydrologic approach outline in the manuscript. I strongly implore the authors consider and describe the synergies with these other ongoing efforts so that all these efforts are supported and elevated.

Thank you very much for highlighting the connection to the social-ecological systems approach. We already cited the original publication of Huggins et al. (2023) in Groundwater that outlined the underlying ideas of the follow-up article that you are referring to. "Groundwater is connected to the water cycle and social, economic, and ecological systems (Huggins et al., 2023)." The ERL article that the reviewer is referring to is currently still under review, but we are happy to include it here and iterate on concrete data products if it is published in time.

Again, we would like to emphasize that interlinkages with other sectors are portrayed as an opportunity and a reason why the Groundwater sector contributes an important component to the ISIMIP experiments; however, these interlinkages are not the primary focus of the sector, especially not in the early phase of establishing a new sector.

Still, we see this comment as an opportunity to highlight that the Groundwater sector can be a catalyst for holistic cross-sectoral modeling and added this last sentence to the end of the paper: "With its various connections to other sectors, the Groundwater sector can be a catalyst for developing new holistic cross-sector modelling efforts that account for the multitude of interconnections between the water cycle and social, economic, and ecological systems."

Overall, I am unsure it makes sense to consider or brand this effort as an ISIMIP 'sector'. My understanding is that in the context of ISIMIP, a "sector" refers to a thematic area of climate impact modeling that groups together models and research focused on a particular domain of human or natural systems affected by climate change. These sectors are broad like Agriculture and Forestry and not really specific components of the water cycle like 'groundwater'. I suggest the authors consider this framing and whether it is consistent with ISIMIP more broadly. Should groundwater really be treated as a sub-component or cross-sectoral area?

We thank the reviewer for this critical comment, as this points to longstanding issues at the core of model intercomparisons. The reason modeling groups chose to focus on different compartments or scales is that the scientific area of "Water" is too broad to be adequately assessed with a single model (equally, Agriculture and Forestry could also be part of a Land cover or Plant sector). Ultimately, we hope to develop a holistic understanding of the water cycle, but this may necessitate building models along the way that address specific research questions and are "simple" enough to be understood (e.g., "parsimonious" to a certain extent). In the end, from the perspective of a model intercomparison, it boils down to having models that can be compared, i.e., having models that can handle the same forcings and can produce the same output variables. This non-trivial selection process then also governs which groups of modeling teams agree on a set of experiments they are willing to conduct to compare model outputs. While others in the community have also expressed that Groundwater, Quality, Global and Regional water, along with Lakes, could be considered one sector, the differences between models are too significant to permit a joint sector.

Thus, the ISIMIP sectors can be broad, such as in "Agriculture", but also more specific, as in the "Regional Forests" or "Lake" sector. This is also the reason why the Groundwater Sector is already accepted by ISIMIP as a separate sector: https://www.isimip.org/about/#sectors-and-contacts. As coordinators of ISIMIP are part of the author team, we know that our sector aligns well with the scope of ISIMIP.

To ensure it is transparent to the reader that the creation of this sector is consistent with other sectors and is driven by the existence of different models, we added additional description to section 2. We also agree that groundwater is not isolated and that there is overlap with the (global) water sector. But this point can be made (to a more or lesser extent) for all components of the Earth system, which are never truly isolated. We still often decide to focus on subcomponents for practical and scientific purposes. We explain our reasoning in some more detail in our revised manuscript.

Line 143 now reads:

*"The creation of a new ISIMIP Groundwater sector is not linked to any funding and is a community-driven effort that includes all modeling groups that wish to participate. During the creation process, multiple groups and institutions were contacted to participate, and additional modeling groups are welcome to join the sector in the future. Models participating in the sectors do not need to be able to model all variables and scenarios defined in the protocol. ISIMIP sectors can be linked to broader thematic concepts, such as Agriculture, or can focus on specific components of the Earth system, such as Lakes or Groundwater (see also https://www.isimip.org/about/#sectors-and-contacts). The separation into these sectors is driven by the availability of models that can be integrated into a model-intercomparison framework, which is based on the same climatic and human forcings and produces a set of comparable output variables. We would like to note that groundwater is not an isolated system, but rather part of the water cycle and the Earth system as a whole. Focusing on it within a dedicated sector aligns well with the existing models and is useful for studying groundwater systems in a thematically focused way. Collaboration (and perhaps integration) with sectors like the Global Water sector is possible and desirable in the future. We discuss possible future synergies with other existing ISIMIP sectors in Section 5."*

On a related note, I was also confused about what all the things around the outside of Figure 4 are… Is agro-economic modeling really a sector in ISIMIP?

Yes, please refer to https://www.isimip.org/about/#sectors-and-contacts for a full list and description of sectors. We also added this link to the sector description in section 2.

I think the authors could do much more work to make Figure 4 more useful… what are the linkages that are really? how would they be developed? what models would you use? how could this be improved by better incorporating the initiatives mentioned above?

The interlinkages between Groundwater and other sectors within ISIMIP are potentially very large and are not only limited by our process understanding (i.e., where groundwater matters), but also by the models that participate in the sectors and their capability to utilize an output variable from another sector as input (i.e., because greater model modifications are necessary and groups lack the resources to implement that change). While papers that outline the interconnection of groundwater help support pursuing these interconnections, the realization will depend significantly on the availability of resources to develop protocols through joint workshops and model implementation changes.

Thus, the actual realization of the interlinkage potential may differ significantly between sectors and within sectors between models. Therefore, at this point, we can only highlight the potential that shows the impact the Groundwater sector can have for a more holistic integration of sectors within ISIMIP. To provide a tangible pathway forward, we have selected a subset of sectors with whom we aim to target closer collaboration in the short term to develop interconnections more closely (green and orange arrows in the previous version of the manuscript).

To improve our figure, we added concrete variables that could be transferred between sectors as a more concrete starting point for future model development:

[Figure]

**Figure 4**: The Groundwater sector provides the potential for multiple interlinkages between different sectors within ISIMIP. In the coming years, we will focus on links to three sectors (green and orange): Water (global), Water (regional), and Water Quality. Other cross-sectoral linkages between non-Groundwater sectors (i.e., linkages between the outer circle) are not shown. Sectors that are currently under development or have not yet have data or outputs that could be shared or used for cross-sectoral assessments are shown in gray. Interactions between sectors are annotated with example processes, key variables, or datasets that can be shared between sectors.

I was also surprised to see that PCR GLOB-WB was not mentioned or included eventhough it has been important to a number of global groundwater studies. I would clarify the recent for this.

The modeling group was approached, but did not find the time to participate in the sector yet. They can always join later if they wish to do so. However, the work of the VIC-Wur model by Inge de Graaf (co-author and one of the coordinators of the Groundwater sector) is closely related to the PCR model, as it is based on her earlier work in implementing the groundwater model in PCR.

Based on the review criteria of GMD....

Scientific significance: Fair (3)

Scientific quality: Poor (4)

Scientific reproducibility: N/A

Presentation quality: Fair (3)

Overall, I think I would focus the article on the idea of the ISIMIP groundwater 'sector' and drop section 4 since it seems scientifically inadequate as is, and significantly deepen the discussion and analysis.

We thank the reviewer for the critical comments, which motivated us to clarify several aspects in our manuscript. We have clarified the motivation for the ISIMIP Groundwater sector and the setup of ISIMIP. Due to the positive comments of Reviewer 2 and their request to expand this section, we chose to keep section 4 and deepen the discussion, thereby also addressing the concerns of Reviewer 1. Even if it is only an exploratory picture of the sector and not a comprehensive analysis, it provides ensemble outputs that have never been combined before and offers an important overview of what the sector can produce in the future. We clarified this in a revised version of our manuscript and also improved on this section as requested by Reviewer 2.

[revised manuscript text omitted]

---

## Referee Report (RR1)

Dear Thomas Wild,

Thank you very much for handling our manuscript. Below, we respond to the two reviewers' feedback. As a result, we have improved the manuscript in multiple ways:

- We clarified the motivation of our sector and the community aspect of the creation of the sector.
- We expanded on possible challenges in the conclusions.
- We expanded the discussion of interlinkages to other sectors and improved Figure 4.
- And we improved on the discussion of the results and conclusions we drew.

In the spirit of a community project, we also added two additional authors: Wim Thiery and Tanjila Akhter, because they contributed to the conceptualization of the sector in previous workshops and this revision. Moreover, they are committed to providing model outputs to the ISIMIP Groundwater sector, utilizing the CLM model in their respective groups.

Below, we provide our responses to the comments in blue. To ensure a more streamlined integration of feedback from both reviewers, we chose to address reviewer 2's feedback first, followed by reviewer 1's.

On behalf of all authors,

Robert Reinecke

**Reviewer 2**

This paper provides an overview of the recently established groundwater sector within the Inter Sectoral Impact Modeling Intercomparison Project (ISIMIP). The paper is quite succinct, and for the most part, strikes a good balance of keeping the presentation high level while still being informative. A small amount of analysis is provided in Section 4 to offer an initial glimpse at notable differences in model outputs for water table depth and recharge, but the aim of this paper is to introduce the motivation for groundwater ISIMIP sector, some background on the participating models, and the sector's short to medium term vision. Overall, I think this paper provides a very approachable and well-presented overview of the new groundwater sector. I have some minor comments that could improve the manuscript, but no major criticisms that need to be addressed.

We thank the reviewer for taking the time to read our manuscript and appreciate the constructive feedback.

Minor comments:

- Section 6 could be improved by adding a few sentences about some potential challenges, technical, logistical, monetary (funding), for realizing the goals of new Groundwater MIP sector.

We have expanded section 6 with the following (line 384):

*"In summary, the ISIMIP Groundwater sector aims to enhance our understanding of the impacts of climate change and direct human impacts on groundwater and a range of related sectors. To realize this goal, the new ISIMIP Groundwater sector will address numerous challenges. For instance, core simulated variables, such as water table depth and recharge, are highly uncertain and difficult to compare with observations. Further, tracing down explanations for inter-model differences will require the joint development and application of new evaluation methods (Eyring et al., 2016b) and protocols. Currently, models of the Groundwater sector operate at different spatial resolutions, and compared to other sectors, they often run at relatively high spatial resolutions, which will need to be addressed in evaluation and analysis approaches. Furthermore, depending on the model, executing single-model simulations already requires substantial amounts of computation time, and running all impact scenarios may be infeasible for some modeling groups. Lastly, running simulations for ISIMIP requires not only computational resources but also human resources, which might not be feasible for all groups. This has always been the case with ISIMIP, and it is an issue that other sectors have faced as well. Still, we are confident that the groundwater sector will enhance our understanding of groundwater within the Earth system and help to promote dialogue and synthesis in the research community. With its various connections to other sectors, the Groundwater sector can be a catalyst for developing new holistic cross-sector modelling efforts that account for the multitude of interconnections between the water cycle and social, economic, and ecological systems."*

- Something not touched on is that certain models could perform better in specific regions on for specific output variables. There could be value in more directly stating that the Groundwater ISIMIP sector could inform region-specific model recommendations for specific outputs.

Thank you for this suggestion. We have added this aspect to section 6.

Line 364 now reads:

*"Model intercomparison and validation may also help identify models that perform better in specific regions or for specific output variables, thus enabling the provision of region- or variable-specific recommendations and uncertainty assessments to subsequent data users."*

- L 135: could the authors expand slightly on "functional relationships"?

We agree that the current explanation may be too brief. We have expanded on this as requested.

Line 155 now reads:

*"We aim to utilize these simulations for an in-depth model comparison, including a comparison to observational data such as time series of groundwater table depth (e.g., Jasechko et al. (2024)) and by utilizing so-called functional relationships (Reinecke et al., 2024; Gnann et al. 2023). Functional relationships can be defined as covariations of variables across space and/or time, and they are a key aspect of our theoretical knowledge of Earth's functioning. Examples include relationships between precipitation and groundwater recharge (Gnann et al. 2023; Berghuijs et al. 2024) or between topographic slope and water table depth (Reinecke et al., 2024)."*

- L 136-138 Groundwater is new to ISIMIP, but are there any notable previous groundwater model intercomparison efforts worth mentioning, even if not global in scale?

We are not aware of an intercomparison project that utilizes multiple groundwater models, especially for large-scale analyses. There are examples of benchmarking experiments that use different groundwater modeling approaches for the same problem or run the same model for different scenarios, but we are not aware of an effort, even on regional scales, that has approached such an intercomparison. We have added this discussion to the introduction and also added a reference to GroMoPo as requested by Reviewer 1.

Line 106 in the introduction now reads:

"[...] *The new Groundwater sector is a separate but complementary sector to the existing global water sector. To our knowledge, there are currently no long-term community efforts for a structured model intercomparison project for groundwater models. While studies have benchmarked different model approaches (e.g., Maxwell et al. 2014), or compared model outputs (Reinecke et al., 2021; 2024), or collected information on where and how we model groundwater (Telteu et al., 2021; Zipper et al., 2023; Zamrsky et al., 2025), no effort yet aims at forcing different groundwater models with the same climate and human forcings for different scenarios.*"

- L179-189: In addition to using the same forcing data, are there plans to run these models at the same resolution or are outputs going to be scaled to the same resolution as a post-processing step?

In the short term, there are no plans to run the models at the same resolutions, as this might require substantial changes in some models or may not even be possible for others because of computation time and resource limitations.  Model outputs are scaled to the same resolution as a post-processing step, which makes it more feasible to conduct model intercomparisons. In this way, more groundwater models are likely to participate. However, some of the models already run on the same resolution or are flexible in running on multiple spatial and temporal resolutions – it would be worth investigating the differences in scale sensitivity.

As the sector further develops in the future, it may become possible and desirable to harmonize spatial resolution for improved inter-comparability of model outputs.

We now clarify this in section 3 (line 190):

"The current sector protocol defines a targeted spatial resolution of 5 arcmin, as this represents not only the resolution achievable by most global models but also the coarsest resolution at which meaningful representation of groundwater dynamics, particularly lateral groundwater flows and water table depths, can still be captured (Gleeson et al., 2021).  ISIMIP3 also specifies experiments with different spatial resolutions, but whether this is achievable with a sub-ensemble of the presented models remains unclear, as it depends on the available computational time, flexibility of model setups, and data availability. To ensure consistency and comparability, the model outputs are currently post-processed by the modeling groups to aggregate their outputs to the protocol-specified spatial and temporal resolutions."

- Figure 1 & Lines 195-196: "This difference in ensemble WTD points to conceptual differences between the models, which should be investigated further." Given the authors on this paper have expertise with these models, could there be a couple

sentences offered positing as to why there could be such stark differences in the water depth for G3M & CLM compared to WBM and VIC-wur?

We agree that a more in-depth discussion of these differences is necessary, which is a key motivation for future research. However, it is also challenging to pinpoint precisely why the presented models differ so much without a much more thorough analysis. To provide better guidance for subsequent studies, we tried to expand on this aspect as much as possible. Line 234 now read:

*"This difference in ensemble WTD points to conceptual differences between the models. $G^3M$ and CLM both use the relatively shallow WTD estimates of Fan et al. (2013) as initial state or spin-up, which could explain the overall shallow water table depth. The difference between $G^3M$ and VIC-wur is consistent with the findings in Reinecke et al. (2024), which showed a deeper water table simulated by the de Graaf et al. (2017) groundwater model, which developed an aquifer parameterization adapted and conceptually similar to VIC-wur and WBM. This difference may be linked to the implementation of groundwater drainage/surface water infiltration or transmissivity parameterizations (Reinecke et al., 2024) as well as differences in groundwater recharge (Reinecke et al., 2021). Furthermore, the models are not yet driven by the same climatic and human forcings, thereby possibly causing different model responses. The newly initiated ISIMIP Groundwater sector offers an opportunity to investigate these differences much more systematically in future studies, for example, by ruling out forcing as a driver of the model differences and by exploring spatial and temporal relationships with key groundwater drivers such as topography (e.g., Reinecke et al., 2024). In addition, the ISIMIP Groundwater sector provides a platform for using the modelling team's expertise on their model implementations (e.g., model structures and parameter fields) to better understand the origins of these differences."*

- Figures 2 and 3: It is my understanding that for this initial comparison the groundwater recharge model results have different forcings (Table A1). I think it would be good to remind the audience of this because the rest of the paper is focused on the forthcoming efforts using the *same* forcings for the MIP.

Thank you for this suggestion. We have added the following sentence to all figures: "Models shown are not yet driven by the same meteorological forcing (see also table A1)."

Minor spelling comments:

- Spelling L150: "and surface water exchange fluxes as upper boundary conditionals without later fluxes": Later**al** missing "al"

Thank you for spotting this, we corrected it.

- Line 275: ISMIP missing I, ISIMIP

Thank you for spotting this, we corrected it.

**Reviewer 1**

I appreciate the efforts of Reinecke et al, and support the effort to better represent groundwater in ISIMIP - this is a much needed, and long called for effort. But overall this manuscript feels a

thin, uncritical, non-exhaustive and somewhat repetitive. This may strongly worded, but it feels more like a paper written quickly after a great workshop rather than a deep effort with longer rumination and iteration. The manuscript seems thin in that each section seems quick and brief rather than deeply insightful or critical. I think a number of the ideas could be expanded upon with more critique and reflection. For example, when I look at the models in Table 2 compared to the linkages in Figure 4, I was struck by the limited capacity of most models to simulate outputs that would be useful for other sectors. At a basic level, if water use is not even in a model, how is it useful to assess water resources? And nothing to do with groundwater quality or contamination is mentioned in Table 1 so how can this effort be useful for water quality?

Deciding on a multi-model intercomparison protocol is not a trivial task. We thank the reviewer for the thoughtful feedback and for taking the time to read our manuscript.[1] However, we would also like to point out that the assessment of the paper quality stands in contrast to the evaluation of Reviewer 2: *"This paper provides an overview of the recently established groundwater sector within the Inter Sectoral Impact Modeling Intercomparison Project (ISIMIP). The paper is quite succinct, and for the most part, strikes a good balance of keeping the presentation high level while still being informative."*

We appreciate that the reviewer recognizes the success of our workshops; however, we would like to clarify that the paper is the outcome of multiple in-person workshops, held in Potsdam (2022, 2025), Prague (2023), Mainz (2024), as well as a EGU splinter meeting (2024) and numerous uncounted online meetings.

We believe that some of the comments stem from a lack of explanation for the motivation behind ISIMIP and the newly created sector. The critical comments thus helped to improve the manuscript. To improve our manuscript, we have in particular clarified the motivation behind our sector and the community aspect of its creation, expanded the discussion of interlinkages to other sectors, improved Figure 4, and expanded on the discussion of model differences.

To specifically address "if water use is not even in a model, how is it useful to assess water resources? And nothing to do with groundwater quality or contamination is mentioned in Table 1 so how can this effort be useful for water quality?"

Water uses are included in some models but not in all models. Furthermore, to better understand water resources, it is essential to comprehend the processes and interactions within the water cycle.[2] This better understanding does not necessarily involve only the 'scenario' of water uses. ISIMIP also defines scenarios of "naturalized" conditions without human intervention. Some models can simulate multiple scenarios, while others may not.

[3] Water quality is omitted here because global groundwater quality data remain sparse. Even surface water quality studies at this scale are nascent, prompting initiatives like ISIMIP's new water quality sector (see https://www.isimip.org/about/#sectors-and-contacts) to address this gap (with currently a paper on its protocol in the second round of review with ERL-Water).

We, however, acknowledge that scientists less familiar with or not involved in the ISIMIP setup may need further explanation on how community decision-making informs the creation of a sector and potential connections to other sectors. We have also expanded the description of ISIMIP to clarify the independence of sectors and models.

Line 139 now reads:

Number: 1 Author:     Subject: Comment on Text     Date: 11/21/25, 11:46:56 AM

Weak argument. Best not to base scientific arguments on 'whataboutisms'.

Number: 2 Author:     Subject: Comment on Text     Date: 11/24/25, 11:17:20 AM

Models without human forcings (mainly pumping) tend to be radically different than models including it, except in basins with small human demand for groundwater. In that case, there is the danger of blythly comparing apples and oranges.

Number: 3 Author:     Subject: Comment on Text     Date: 11/21/25, 11:55:52 AM

Yes, but I would also add that the first step toward groundwater quality modeling is groundwater quantity (and flux) modeling. So the advocated work can be viewed as one step in that direction.

*"The creation of a new ISIMIP Groundwater sector is not linked to any funding and is a community-driven effort that includes all modeling groups that wish to participate. During the creation process, multiple groups and institutions were contacted to participate, and additional modeling groups are welcome to join the sector in the future. Models participating in the sectors do not need to be able to model all variables and scenarios defined in the protocol. ISIMIP sectors can be linked to broader thematic concepts, such as Agriculture, or can focus on specific components of the Earth system, such as Lakes or Groundwater. We would like to note that groundwater is not an isolated system, but rather part of the water cycle and the Earth system as a whole. Focusing on it within a dedicated sector aligns well with the existing models and is useful for studying groundwater systems in a thematically focused way. Collaboration (and perhaps integration) with sectors like the Global Water sector is possible and desirable in the future. We discuss possible future synergies with other existing ISIMIP sectors in Section 5."*

We also clarified that the usage of outputs for other sectors depends on the scientific question and specified how the quality sector might use outputs from the Groundwater sector in the future in section 5 (line 314):

*"Furthermore, the relevance of groundwater for water quality assessments is widely recognized (e.g., for phosphorous transport from groundwater to surface water (Holman et al., 2008), or for salinization (Kretschmer et al., 2025), or as a link between warming groundwater and stream temperatures (Benz et al., 2024). And the community effort of Friends of Groundwater called for a global assessment of groundwater quality (Misstear et al., 2021). The Water Quality sector could incorporate model outputs from the Groundwater sector as input to improve, for example, their estimates of groundwater contributions to surface water quantity or leakage of surface water to groundwater. On the other hand, the Groundwater sector can utilize estimates of the Water Quality sector to better assess water availability by incorporating water quality criteria. Ultimately, this may also result in advanced groundwater models in the Groundwater sector that account for quality-related processes directly, which can then be integrated into a future modeling protocol. One of the models ($G^3M$; see Table 1) is already capable of simulating salinization processes.*

[1] *Leveraging such connections between sectors will provide valuable insights beyond groundwater itself. The outputs and models that can be used for intersectoral assessments depend on the research question and may necessitate the use of only a subset of models from an ensemble. Specifically, considering groundwater quality, a collaboration between both sectors could be achieved in multiple aspects. Integrating groundwater availability with water quality helps ensure sufficient and safe drinking and irrigation water. Focusing on aquifer storage levels and pollutant loads can help maintain groundwater resilience, safeguard food security, and protect public health under changing climate and socioeconomic conditions. Further, integrating groundwater quantity data with pollution source mapping helps prioritize remediation efforts where aquifers are most vulnerable, ensuring both water availability and quality. Concerning observational data, a unified approach to collecting and developing shared databases for groundwater levels and water quality measurements across multiple agencies reduces bureaucratic hurdles and ensures consistent, comparable data. Using standardized procedures for dealing with observational uncertainties, such as data gaps, scaling issues, and measurement inconsistencies, would support collaborative research further."*

Number: 1 Author:     Subject: Comment on Text     Date: 11/21/25, 12:04:06 PM

OK, but the authors do not seem to be aware of the especially stubborn challenges of modeling regional groundwater quality -- a task that dramatically more difficult than modeling regional groundwater quantity. See for example:

Fogg, G. E., & LaBolle, E. M. (2006). Motivation of synthesis, with an example on groundwater quality sustainability: MOTIVATION OF SYNTHESIS. *Water Resources Research*, *42*(3). https://doi.org/10.1029/2005WR004372

Section 4 about unstructured experiments seemed repetitive to other recent articles on uncertainty in the water table depth and recharge including those of co-authors. It also felt thin and preliminary, and frankly uninspiring (in that the models seem to show little consistency) and unsurprising (due to overlap with previous articles).

We thank the reviewer for this comment. It is true that this section provides only limited new insights, but this was not the primary aim here, as stated in the original text: *"We opted for a straightforward initial comparison due to the various data formats, model resolutions, and forcings that complicate a more thorough examination of a specific scientific inquiry. Thus, this descriptive analysis serves as an introductory overview that highlights the present state of the art and identifies model discrepancies warranting further investigation."*

To clarify this further, we added "One of our goals in the Groundwater sector is to conduct extensive analysis to better illustrate and understand the model differences. The analysis presented here is intended solely as an introductory overview to provide a sense of the rationale behind our initiative."

In response to Reviewer 2, we also improved the discussion of the model differences.

Line 234 now read:

*"This difference in ensemble WTD points to conceptual differences between the models. $G^3M$ and CLM both use the relatively shallow WTD estimates of Fan et al. (2013) as initial state or spin-up, which could explain the overall shallow water table depth. The difference between $G^3M$ and VIC-wur is consistent with the findings in Reinecke et al. (2024), which showed a deeper water table simulated by the de Graaf et al. (2017) groundwater model, which developed an aquifer parameterization adapted and conceptually similar to VIC-wur and WBM. This difference may be linked to the implementation of groundwater drainage/surface water infiltration or transmissivity parameterizations (Reinecke et al., 2024) as well as differences in groundwater recharge (Reinecke et al., 2021). The newly initiated ISIMIP Groundwater sector offers an opportunity to investigate these differences much more systematically in future studies, for example, by ruling out forcing as a driver of the model differences and by exploring spatial and temporal relationships with key groundwater drivers such as topography (e.g., Reinecke et al., 2024). In addition, the ISIMIP Groundwater sector provides a platform for using the modelling team's expertise on their model implementations (e.g., model structures and parameter fields) to better understand the origins of these differences."*

In addition, we would like to highlight two novel aspects that differ from the previously published results of the co-authors. Here, we present a different set of models involved compared to Reinecke et al. (2024) (specifically WBM and CLM, and in part also VIC-Wur even if it is conceptually similar to the de Graaf model discussed in Reinecke et al. 2024), and we include models that specifically incorporate processes not included in previous assessments by Reinecke et al., such as karst. We thus also added the following statement in line 219:

*"Some overlap with recent model comparison studies naturally exists (e.g., Gnann et al., 2023; Reinecke et al., 2024, Reinecke et al. 2021), even though this brief analysis contains a different ensemble of models and thus provides new insights."*

Examples of it not being exhaustive is that it does not even mention the recent GroMoPo effort that a number of the authors have been involved with (Zipper et al. 2023; Zamrsky et al., 2025).

This initiative has compiled hundreds of regional scale model even though line 98 claims to 'integrate currently available groundwater models that operate at regional scale'.

We thank the reviewer for the suggestion to cite the GroMoPo project. Importantly, while GroMoPo collected information about groundwater models, it did not collect the models themselves or any knowledge about how to operate them. Thus, they do not provide any basis for being used in a model ensemble. The ISIMIP sector, of course, is open to any modeling groups that would still like to join the intercomparison initiative. We agree, however, that the phrasing in line 98 can be misleading, as the current sector integrates models of modeling teams that decided to commit their time to joint experiments.

We now cite GroMoPo in the introduction and specify why it is not directly helpful for the creation of this sector. Line 107 now reads:

*"To our knowledge, there are currently no long-term community efforts for a structured model intercomparison project for groundwater models. While studies have benchmarked different modeling approaches (e.g., Maxwell et al. 2014), compared model outputs (Reinecke et al., 2021; 2024), or collected information on where and how we model groundwater (Telteu et al., 2021; Zipper et al., 2023; Zamrsky et al., 2025), no effort yet aims at forcing different groundwater models with the same climate and human forcings for different scenarios."*

And we have rephrased the initial sentence of the paragraph to be more specific:

Line 100 now reads:

*"Here, we present a new sector in ISIMIP called the ISIMIP Groundwater Sector, which integrates models of the groundwater community that operate at regional (at least multiple km$^2$ (Gleeson and Paszkowski, 2014)) to global scales and are committed to providing model simulations to this new sector."*

Also missing are any mention of linking with global groundwater quality and contamination efforts such as Friends of Groundwater which seems important for the groundwater quality linkage.

Thank you for this suggestion. We are now citing the Friends of Groundwater initiative in line 317:

*"Furthermore, the relevance of groundwater for water quality assessments is widely recognized (e.g., for phosphorous transport from groundwater to surface water (Holman et al., 2008), or for salinization (Kretschmer et al., 2025), or as a link between warming groundwater and stream temperatures (Benz et al., 2024). And the community effort of Friends of Groundwater called for a global assessment of groundwater quality (Misstear et al., 2021). The Water Quality sector could incorporate model outputs from the Groundwater sector as input to improve, for example, their estimates of groundwater contributions to surface water quantity or leakage of surface water to groundwater."*

Finally, I was a recent reviewer of this manuscript by Huggins et al. (again with some of the same coauthors) and am struck that many of the linkages to other sectors would be much better created by taking a more holistic, social-ecological systems approach or at least bringing in insights and data from this approach than the narrow hydrologic approach outline in the manuscript. I strongly implore the authors consider and describe the synergies with these other ongoing efforts so that all these efforts are supported and elevated.

Thank you very much for highlighting the connection to the social-ecological systems approach. We already cited the original publication of Huggins et al. (2023) in Groundwater that outlined the underlying ideas of the follow-up article that you are referring to. "Groundwater is connected to the water cycle and social, economic, and ecological systems (Huggins et al., 2023)." The ERL article that the reviewer is referring to is currently still under review, but we are happy to include it here and iterate on concrete data products if it is published in time.

Again, we would like to emphasize that interlinkages with other sectors are portrayed as an opportunity and a reason why the Groundwater sector contributes an important component to the ISIMIP experiments; however, these interlinkages are not the primary focus of the sector, especially not in the early phase of establishing a new sector.

Still, we see this comment as an opportunity to highlight that the Groundwater sector can be a catalyst for holistic cross-sectoral modeling and added this last sentence to the end of the paper: "With its various connections to other sectors, the Groundwater sector can be a catalyst for developing new holistic cross-sector modelling efforts that account for the multitude of interconnections between the water cycle and social, economic, and ecological systems."

Overall, I am unsure it makes sense to consider or brand this effort as an ISIMIP 'sector'. My understanding is that in the context of ISIMIP, a "sector" refers to a thematic area of climate impact modeling that groups together models and research focused on a particular domain of human or natural systems affected by climate change. These sectors are broad like Agriculture and Forestry and not really specific components of the water cycle like 'groundwater'. I suggest the authors consider this framing and whether it is consistent with ISIMIP more broadly. Should groundwater really be treated as a sub-component or cross-sectoral area?

We thank the reviewer for this critical comment, as this points to longstanding issues at the core of model intercomparisons. The reason modeling groups chose to focus on different compartments or scales is that the scientific area of "Water" is too broad to be adequately assessed with a single model (equally, Agriculture and Forestry could also be part of a Land cover or Plant sector). Ultimately, we hope to develop a holistic understanding of the water cycle, but this may necessitate building models along the way that address specific research questions and are "simple" enough to be understood (e.g., "parsimonious" to a certain extent). In the end, from the perspective of a model intercomparison, it boils down to having models that can be compared, i.e., having models that can handle the same forcings and can produce the same output variables. This non-trivial selection process then also governs which groups of modeling teams agree on a set of experiments they are willing to conduct to compare model outputs. While others in the community have also expressed that Groundwater, Quality, Global and Regional water, along with Lakes, could be considered one sector, the differences between models are too significant to permit a joint sector.

Thus, the ISIMIP sectors can be broad, such as in "Agriculture", but also more specific, as in the "Regional Forests" or "Lake" sector. This is also the reason why the Groundwater Sector is already accepted by ISIMIP as a separate sector: https://www.isimip.org/about/#sectors-and-contacts. As coordinators of ISIMIP are part of the author team, we know that our sector aligns well with the scope of ISIMIP.

To ensure it is transparent to the reader that the creation of this sector is consistent with other sectors and is driven by the existence of different models, we added additional description to

section 2. We also agree that groundwater is not isolated and that there is overlap with the (global) water sector. But this point can be made (to a more or lesser extent) for all components of the Earth system, which are never truly isolated. We still often decide to focus on subcomponents for practical and scientific purposes. We explain our reasoning in some more detail in our revised manuscript.

Line 143 now reads:

*"The creation of a new ISIMIP Groundwater sector is not linked to any funding and is a community-driven effort that includes all modeling groups that wish to participate. During the creation process, multiple groups and institutions were contacted to participate, and additional modeling groups are welcome to join the sector in the future. Models participating in the sectors do not need to be able to model all variables and scenarios defined in the protocol. ISIMIP sectors can be linked to broader thematic concepts, such as Agriculture, or can focus on specific components of the Earth system, such as Lakes or Groundwater (see also https://www.isimip.org/about/#sectors-and-contacts). The separation into these sectors is driven by the availability of models that can be integrated into a model-intercomparison framework, which is based on the same climatic and human forcings and produces a set of comparable output variables. We would like to note that groundwater is not an isolated system, but rather part of the water cycle and the Earth system as a whole. Focusing on it within a dedicated sector aligns well with the existing models and is useful for studying groundwater systems in a thematically focused way. Collaboration (and perhaps integration) with sectors like the Global Water sector is possible and desirable in the future. We discuss possible future synergies with other existing ISIMIP sectors in Section 5."*

On a related note, I was also confused about what all the things around the outside of Figure 4 are... Is agro-economic modeling really a sector in ISIMIP?

Yes, please refer to https://www.isimip.org/about/#sectors-and-contacts for a full list and description of sectors. We also added this link to the sector description in section 2.

I think the authors could do much more work to make Figure 4 more useful... what are the linkages that are really? how would they be developed? what models would you use? how could this be improved by better incorporating the initiatives mentioned above?

The interlinkages between Groundwater and other sectors within ISIMIP are potentially very large and are not only limited by our process understanding (i.e., where groundwater matters), but also by the models that participate in the sectors and their capability to utilize an output variable from another sector as input (i.e., because greater model modifications are necessary and groups lack the resources to implement that change). While papers that outline the interconnection of groundwater help support pursuing these interconnections, the realization will depend significantly on the availability of resources to develop protocols through joint workshops and model implementation changes.

Thus, the actual realization of the interlinkage potential may differ significantly between sectors and within sectors between models. Therefore, at this point, we can only highlight the potential that shows the impact the Groundwater sector can have for a more holistic integration of sectors within ISIMIP. To provide a tangible pathway forward, we have selected a subset of sectors with whom we aim to target closer collaboration in the short term to develop interconnections more closely (green and orange arrows in the previous version of the manuscript).

To improve our figure, we added concrete variables that could be transferred between sectors as a more concrete starting point for future model development:

[Figure]

**Figure 4**: The Groundwater sector provides the potential for multiple interlinkages between different sectors within ISIMIP. In the coming years, we will focus on links to three sectors (green and orange): Water (global), Water (regional), and Water Quality. Other cross-sectoral linkages between non-Groundwater sectors (i.e., linkages between the outer circle) are not shown. Sectors that are currently under development or have not yet have data or outputs that could be shared or used for cross-sectoral assessments are shown in gray. Interactions between sectors are annotated with example processes, key variables, or datasets that can be shared between sectors.

I was also surprised to see that PCR GLOB-WB was not mentioned or included eventhough it has been important to a number of global groundwater studies. I would clarify the recent for this.

The modeling group was approached, but did not find the time to participate in the sector yet. They can always join later if they wish to do so. However, the work of the VIC-Wur model by Inge de Graaf (co-author and one of the coordinators of the Groundwater sector) is closely related to the PCR model, as it is based on her earlier work in implementing the groundwater model in PCR.

Based on the review criteria of GMD....

Scientific significance: Fair (3)

Scientific quality: Poor (4)

Scientific reproducibility: N/A

Presentation quality: Fair (3)

Overall, I think I would focus the article on the idea of the ISIMIP groundwater 'sector' and drop section 4 since it seems scientifically inadequate as is, and significantly deepen the discussion and analysis.

We thank the reviewer for the critical comments, which motivated us to clarify several aspects in our manuscript. We have clarified the motivation for the ISIMIP Groundwater sector and the setup of ISIMIP. Due to the positive comments of Reviewer 2 and their request to expand this section, we chose to keep section 4 and deepen the discussion, thereby also addressing the concerns of Reviewer 1. Even if it is only an exploratory picture of the sector and not a comprehensive analysis, it provides ensemble outputs that have never been combined before and offers an important overview of what the sector can produce in the future. We clarified this in a revised version of our manuscript and also improved on this section as requested by Reviewer 2.

[4]his will yield a new understanding of how these models differ, what the reasons for these differences are, and how they could be improved. In addition, it will provide a basis for implementing impact analyses with ensemble runs based on future scenarios using ISIMIP3b inputs.

**3 The current generation of groundwater models in the sector**

Many large-scale groundwater models are already participating in the sector (Table 1), and we expect it to expand further. The current models are mainly global-scale, with some having a particular regional focus, and primarily using daily timesteps.

While the mainprima[5]y modeling purpose of most models is to simulate parts of the terrestrial water cycle, they all focus on different aspects (such as karst recharge or sea-water intrusion), most investigate interactions between groundwater and land surface processes, and account for human water uses. Two models (V2KARST and GGR) have distinct purposes in modeling groundwater recharge and do not model any head-based groundwater fluxes. Conceptually, the models may be classified according to Condon et al. (2021) into five categories: lumped models with static groundwater configurations of long-term mass balance [6]), saturated groundwater flow with recharge,

**Number: 1 Author:    Subject: Inserted Text         Date: 11/24/25, 12:41:16 PM**
using [every occurrence of "utilize" or "utilizing" in the paper can be replaced with "use" or "using".]

**Number: 2 Author:    Subject: Cross-Out    Date: 11/24/25, 12:41:42 PM**

**Number: 3 Author:    Subject: Highlight    Date: 11/24/25, 12:43:01 PM**
Replace very "groundwater table" with "water table", which is the correct scientific term.

**Number: 4 Author:    Subject: Highlight    Date: 11/24/25, 12:45:01 PM**
No new paragraph. Or, replace "this" with whatever you are referring to.

**Number: 5 Author:    Subject: Cross-Out    Date: 11/24/25, 12:45:29 PM**

**Number: 6 Author:    Subject: Highlight    Date: 11/24/25, 12:50:20 PM**
Would be clearer to put these parentheticals at the beginning of each respective clause.

and surface water exchange fluxes as upper boundary conditions without lateral fluxes (b), quasi 3D models with variably saturated flow in the soil column and a dynamic water table as a lower boundary condition (c), saturated flow models solving mainly the Darcy equation (d), and variably saturated flow which is calculated as three-dimensional flow throughout the entire subsurface below and above the water table (e). See Condon et al. (2021) and also Gleeson et al. (2021) for a more detailed overview and discussion of approaches. Half of the models (Table 1) simulate a saturated subsurface flux (d), while V2KARST and GGR mainly use a 1D vertical approach (b), and others simulate a combination of multiple approaches (ParFlow, Table 1) or can switch between different approaches (CWatM, Table 1).

The sector protocol is defined at https://protocol.isimip.org/#/ISIMIP3a/groundwater and will be updated over time. We have defined multiple joint outputs for this sector (23 variables in total), but not all models can yet provide all outputs (Table 2). Models can provide 1-19 outputs (11 on average), and multiple models have further additional outputs that are currently under development. The global water sector also contains groundwater-related variables (Table A2), enabling groundwater-related analysis. We list them here to show their close connection to the global water sector and facilitate an overview of future groundwater-related studies.

The current sector protocol defines a targeted spatial resolution of 5 arcmin, as this represents not only the resolution achievable by most global models but also the coarsest resolution at which meaningful representation of groundwater dynamics, particularly lateral groundwater flows and water table depths, can still be captured (Gleeson et al., 2021). ISIMIP3 also specifies experiments with different spatial resolutions, but whether this is achievable with a sub-ensemble of the presented models remains unclear, as it depends on the available computational time, flexibility of model setups, and data availability. To ensure consistency and comparability, the model outputs are currently post-processed by the modeling groups to aggregate their outputs to the protocol-specified spatial and temporal resolutions.

Table 1: Summary of all models participating in the ISIMIP Ge groundwater sector. This table lists only models that add new variables to the ISIMIP protocol. Models already part of the global water sector and providing other groundwater-related variables are not listed here. (GMD discussion formatting requires a portrait instead of a landscape table)

| Model name | Main model purpose | Coupling with other models | Spatial domain and resolution | Temporal resolution | Hydrogeo logical configura tion, e.g. number of layers | Conceptu al model according to Condon et al. | Calibrate d | Represen tation of groundw ater use | Main Reference |
|---|---|---|---|---|---|---|---|---|---|
| Water Balance Model (WBM) | Represent ation of the terrestrial hydrologi c cycle, including | - | Global and regional. Spatial resolution defined by the input | Sub-daily, Daily, Multi-day | 1 soil layer, 2 groundwat er layers | d. | Globally: no, regional: yes (NE, US) | Through calculated abstractio ns from groundwat er. | Grogan et al. (2022) With groundwat er methods based on |

**Number: 1 Author: Subject: Highlight Date: 11/24/25, 1:42:34 PM**

Missing from this list is any mention of shallow and deep groundwater phenomena. See Condon et al. 2021 for several important descriptions of why both shallow and deep (e.g., semi-confined and confined systems, where groundwater levels and interplay with surface processes tend to be completely different than the water table) groundwater phenomena and their connections can also be important for adequately representing groundwater storage and impact processes.

[revised manuscript text omitted]

Number: 1 Author:    Subject: Comment on Text    Date: 11/24/25, 2:41:57 PM
The map would be much more informative if there was more color contrast in the 0 to 5 m range. Suggest using a different color gradation.

[Figure]

**Figure 2:** Global groundwater recharge (GWR) in 2001 or at steady-state (only VIC-wur). The simplified boxplot
(a) shows the arithmetic model mean as a colored dot and the median as a black line. Whiskers indicate the 25[th]
and 75[th] percentiles, respectively. The global map (b) shows the coefficient of variation of the model ensemble
without V2KARST. Models shown are not yet driven by the same meteorological forcing (see also table A1).

We further calculated relative changes in groundwater recharge between 2001 and 2006 (Fig. 3) with an ensemble
of 7 models (CLM, CWatM, GGR, VIC-wur, V2KARST, WBM, and ParFlow). The ensemble includes two
models that only simulate specific regions (V2KARST: regions of karstifiable rock, ParFlow: Euro CORDEX
domain). This result shows a potential analysis that should be repeated within the new Geroundwater sector.
Intentionally, we do not investigate model agreement on the sign of change or compare them with observed data.
The ensemble still highlights plausible regions of groundwater recharge changes, such as in Spain and Portugal,
which aligns with droughts in the investigated period (Paneque Salgado and Vargas Molina, 2015; Coll et al.,
2017; Trullenque-Blanco et al., 2024). Relative increases in groundwater recharge are mainly shown for arid
regions in the Sahara, the Middle East, Australia, and Mexico. However, it is likely that because we investigate
relative changes, this might be related to the already low recharge rates in these regions.

Number: 1 Author: Subject: Comment on Text Date: 11/24/25, 2:49:07 PM
The units are clearly wrong. Perhaps it should be "m/yr"? But even that would not look consistent with the ordinate axis units in (a).

[Figure]

a)

b)

Mean relative GWR change 2001 - 2006

-40%    -20%    0    +20%    +40%

[revised manuscript text omitted]

Number: 1 Author: Subject: Comment on Text Date: 11/24/25, 2:57:53 PM

It seems to me that the biggest benefit of the higher fidelity regional groundwater models is to provide 'test beds' for how to upscale processes in the global or continental scale groundwater models. The latter will necessarily neglect or upscale processes in the former, and there should be comparison studies exploring how to adequately represent regional phenomema in the global/continental models.

(fire sector) under climate change are yet to be explored (Fig. 4). To prioritize our efforts and set a research agenda for the groundwater ISIMIP sector, we will first focus on existing and more straightforward connections to the global water sector, regional water sector, and the water quality sector and then expand to collaboration with other sectors (Fig. 4).

**6 A vision for the ISIMIP groundwater sector**

Given groundwater's importance in the Earth system and for society, it is imperative to expand our knowledge of groundwater and (1) how it is impacted by global climate change and other human forcings and 1) how, in turn, this will affect other systems connected to groundwater. This enhanced understanding is essential to equip us with the knowledge needed to address future challenges effectively. The ISIMIP Ggroundwater sector serves as a foundation for examining and measuring the effects of global change on groundwater systems worldwide. It facilitates cross-sector investigations, such as those concerning water quality, examines the influence of various model structures on groundwater dynamics simulations, and supports the collaborative creation of new datasets for model parameterization and assessment.

Already in the short term, the creation of the Ggroundwater sector has substantial potential to enhance large-scale groundwater research by developing better modeling frameworks for reproducible research (running the multitude of experiments targeted in ISIMIP requires an automated modeling pipeline) and forge a community that can critically examine current modeling practices. The simple model comparison presented here sparks first raises initial questions onas to why models differ and invites us to explore model differences in moregreater depth. Such model intercomparison studies will enable us to quantify uncertainties and identify hotspots for model improvement. They will also allowallow us us to assess the impact of climate and land use change on variousdifferent groundwater-related variables, such as groundwater recharge and water table depth, and enableallow ensemble-based impact assessments ofn future water availability. Model intercomparison and validation may also help identify models that perform better in specific regions or for specific output variables, thus allowingenabling the provision of region- or variable-specific recommendations and uncertainty assessments to subsequent data users.

In the long term, the sector will enable us to jointly reflect on processes that we currently do not model or that requireneed improvement, possibly also through new modeling approaches such as hybrid machine-learning models tailored to the large-scale representation of groundwater. These model developments will be incorporatedluded into the groundwater sector's contributions to upcoming ISIMIP simulation rounds, such as

ISIMIP4–, which is scheduledet to commencestart in 2026. Since groundwater is connected to many socio- ecological systems, groundwater models could also emerge as a modular coupling tool that can be integrated into multiple sectors. The newly establishedfounded groundwater sector already provides a first step in that direction by standardizing output names and units. If models are modular enough and define a standardized Application

Programming Interface (API), they could also serve as a valuable tool for other science communities.

The lack of a community-wide coordinated effort to simulate the effects of climate change on groundwater at regional to global scale has precluded the comprehensive consideration of climate change impacts on groundwater

Number: 1 Author:     Subject: Cross-Out     Date: 11/24/25, 3:17:04 PM

Number: 2 Author:     Subject: Inserted Text          Date: 11/24/25, 3:17:30 PM
,

Number: 3 Author:     Subject: Inserted Text          Date: 11/24/25, 3:17:38 PM
,

Number: 4 Author:     Subject: Inserted Text          Date: 11/24/25, 3:16:27 PM

[revised manuscript text omitted]

---

## Author Response (AR2)

Dear Thomas Wild,

Thank you for the extensive review of our revision and the additional reviewer feedback. We made multiple changes to the manuscript with the following major changes:

- We revised the abstract to more closely represent the focus of the paper.
- We uploaded the figure plotting scripts and preprocessing scripts along with the data to allow reproduction of the figures.
- And included additional literature as suggested.

Our responses are highlighted in blue, and text changes in addition are *italicized*.

On behalf of all authors,

Robert Reinecke

**1 Editor comments**

Dear Authors,

Thanks for preparing your thorough revisions in response to the first round of reviewer comments. I am going to pass your manuscript and responses back to reviewers now, but I have a few of my own comments that I would like you to consider addressing at this stage. I have a set of high-level comments, but also some line comments as well.

High-level comments:

1. **I suggest further distinguishing the paper's contribution in the text**. I think I am still struggling to discern this paper's core contribution. I don't mean to suggest it's not a contribution, only that I think further clarifying it could help readers. As far as I can tell, the paper does not deliver any protocol, and it also does not really deliver any results. So, is its purpose ultimately to motivate the value of having the groundwater sector in isimip, and describe some of the models and variables that will be looked at, but without detailed protocols or results given? If so, I would suggest stating this much more explicitly. Also, regarding results, I do like have your section is titled to clarify its purpose "4. Unstructured experiments point out model differences that should be explored further". But earlier in the abstract you refer to 'initial results', which I suggest modifying because I think it could lead to misinterpretation about what this paper actually provides. I do actually agree with reviewer one's original comment that the paper might be better off spending less time showing results from a range of papers/experiments whose points of commonality are hard to discern (section 4), and instead spending more time on the paper's core contribution to introduce the new isimip-groundwater sector, but I respect that you wish to keep this section, and you already provided rationale in your response to reviewers about why you feel it is important to keep.

Thank you for the constructive feedback on our manuscript. Yes, the paper is not delivering results as a core contribution, but the aim is to provide and motivate a modeling protocol. This protocol, however, is embedded in the general ISIMIP protocol and thus less noticeable. We

agree that this needs to be stated more clearly. We also agree that the formulation in the abstract might be unclear. Therefore, we have rephrased the abstract and part of the introduction accordingly. Also based on the comments from Reviewer #1 the abstract now reads:

*Groundwater serves as a crucial freshwater resource for people and ecosystems, playing a vital role in adapting to climate change. Yet, its availability and dynamics are affected by climate variations, changes in land use, and abstraction. Despite its importance, our understanding of how global change will influence groundwater in the future remains limited. Multi-model ensembles are powerful tools for impact assessments; compared to single-model studies, they provide a more comprehensive understanding of uncertainties and enhance the robustness of projections by capturing a range of possible outcomes. However, to date, no ensemble of groundwater models has been available to assess the impacts of global change. Here, we present the new Groundwater sector within ISIMIP, which combines multiple global, continental, and regional-scale groundwater models. We describe the rationale for the sector, the sectoral output variables that underpinned the modeling protocol, and showcase current model differences and possible future analysis. Currently, eight models are participating in this sector, ranging from gradient-based groundwater models to specialized karst recharge models, each producing up to 19 out of 23 modeling protocol-defined output variables. To showcase the benefits of a joint sector, we utilize available model outputs of the participating models to show the substantial differences in estimating water table depth (global arithmetic mean 6 - 127 m) and groundwater recharge (global arithmetic mean 78 - 228 mm/y), which is consistent with recent studies on the uncertainty of groundwater models, but with distinct spatial patterns. We further outline synergies with 13 of the 17 existing ISIMIP sectors and specifically discuss those with the global water and water quality sectors. Finally, this paper outlines a vision for ensemble-based groundwater studies that can contribute to a better understanding of the impacts of climate change, land use change, and abstractions on the world's largest accessible freshwater store – groundwater.*

We extended the introduction with an additional paragraph that summarizes the paper context and thus provides a concrete framing (Line 125 ff):

*"In this manuscript, we provide an overview of the current ISIMIP framework with an emphasis on how the new sector is embedded in the current project in Section 2. The current generation of groundwater models participating in this effort is described and compared, and we define a list of output variables that form the foundation of the sector's model intercomparison protocol in Section 3. In 4, we showcase current model differences and possible future analysis. The connections to other sectors are discussed in Section 5, and Section 6 provides an outlook on future scientific goals for the groundwater sector."*

2. Consistency with 'global change' terminology and literature. The paper's title and exposition invoke 'global change', but I find that this paper is more aligned with climate change and global hydrology, rather than global change. It is true that some of the key models referenced in your study represent dynamics around land use change, climate change scenarios, and human groundwater abstraction. So, in this sense your study is theoretically aligned with global change, assuming you ultimately conduct experiments that explore these dimensions. But, what I think has the potential to create confusion is that there are many researchers who focus centrally on 'global change', and many of

them study or consider water resources in various ways, usually focused on human dimensions, economics, etc. I think it will be unclear to those communities how your contribution aligns (or does not) with their efforts, and it could be helpful clarify this in the paper. For example, as I mention in my line comments below, there are researchers who work on groundwater and global change through a hydro-economics lens, focused foremost on the economics of groundwater extraction, but also including some physics-based representations of groundwater extraction. Many of those teams are also directly linked (and internally consistent) with the IAMs that produce the emissions and land use change trajectories (e.g., via ScenarioMIP) that are responsible for feeding climate impacts information into the hydrology models that you are focused on. How will the groundwater sector in ISIMIP relate to those communities and any global change scenarios they produce related to groundwater?

[moved from below] It is only at the very end of the abstract that we learn that this entire exercise is really focused on climate impacts. I personally think it could lead to misinterpretations (from those not very familiar with ISIMIP) to feature that only at the end of the abstract, particularly given the title's use of "global change".

We specifically chose the title global change because, as the editor notes, we do investigate not only climate change but also land use change, environmental change, and groundwater abstractions that are influenced by human activities on land. Global change is the interplay of societal developments and climate change. We believe that our manuscript touches both aspects. ISIMIP has a strong link to the CMIP and ScenarioMIP community not only being a "downstream" user of the CMIP simulations that are used to drive our models but also a direct feedback link to this community, e.g., the upcoming CMIP7. Having a groundwater sector within ISIMIP perfectly aligns with research communities that already work on the economics of groundwater. And as stated below in a response, we welcome also these kinds of models to the sector if they can be included in a meaningful way. Nevertheless, an interaction to jointly develop new scenarios of e.g. groundwater use would be extremely valuable, especially since some of the models e.g. Superwell (Niazi et al. 2025) is using parameters and inputs that originate in some of the models of the global water and groundwater sector already. In our response below, we specifically refer to economic models now. To clarify that our focus in fact is not only climate change, we have changed the last sentence of the abstract:

*"Finally, this paper outlines a vision for ensemble-based groundwater studies that can contribute to a better understanding of the impacts of climate change, land use change, environmental change, and socio-economic change on the world's largest accessible freshwater store – groundwater."*

3. I don't find that it's possible for me to reproduce the figures/results you present in the paper. Please provide a code repository that complies with GMD standards.

We have added the plotting and preprocessing scripts to the Zenodo repository that was already used in the previous version. We have updated the Zenodo repository with a new version2 and an appropriate DOI. Please note that while the plots can now be reproduced and we provide the preprocessing scripts, we do not republish the original model outputs here, which are listed in Table A1 (as a note, all model runs executed according to the ISIMIP protocol are all publicly available at data.isimip.org).

The data availability statement now reads (lines 419 ff):

*"The ensemble-mean WTD and groundwater recharge trends are available in Reinecke (2025) https://doi.org/10.5281/zenodo.14962511. The Zenodo repository included pre-processing scripts, plotting files, and data, as well as the main outputs presented in this manuscript as raster files. For the original model data publications, see Table A1."*

Line comments (line numbers, when provided, pertain to the tracked changes PDF)

Abstract

- Line 35. I suggest avoiding normative language like "excessive", and instead using more specific language (i.e., excessive in what way), if possible, to avoid misinterpretation.

*It now reads: "Yet, its availability and dynamics are affected by climate variations, changes in land use, and abstraction."*

- Line 38. "no ensemble of groundwater models has been available". I think it would be helpful to articulate how you define a groundwater model for purposes of this paper. I assume you are referring to models focused largely on the physical system. But, for example, there are global hydro-economic models that deal explicitly with the economics of groundwater extraction along with some groundwater physics. Are those models excluded? I assume so, because they are not cited.

[moved from below] Sections 2/3: ISIMIP framework/current generation of models

- Here could be a place to consider describing what kinds of models are included versus excluded—like hydroeconomic models and global change models that are more focused on humans and less focused on groundwater physics.

If these models also produce a subset of output variables of interest to ISIMIP, they would be more than welcome to participate in the future, especially since some of these models already produce overlapping output variables such as abstraction or recharge. Also, if the modeling groups think there would be a benefit of extending the definition of output variables, they would be more than welcome to participate in this sector or create new sectors. A participation would also offer a benefit in utilizing intersectoral synergies with the agro-economic sector (which is currently still under development in ISIMIP and thus also shown in gray in Fig. 4). Still, this does not change the fact that no ensemble of groundwater models is yet available to assess global change impacts. We already stated that in section 2 of the manuscript "[..] and additional modeling groups are welcome to join the sector in the future. Models participating in the sectors do not need to be able to model all variables and scenarios defined in the protocol." We don't think such a definition necessarily belongs in the abstract – especially since we do not restrict a specific type of model. Still, we have changed the introduction to clearly state which current models we consider part of this sector at this phase of the project, and to offer a concrete invitation to others to join this effort.

Line 117 now reads: *"The new sector welcomes all models that are relevant to assessing the impacts of global change on groundwater-related variables. While the current set of models presented here focuses on different physical representations of groundwater, future*

*developments could also include models that account for hydro-economic aspects of groundwater (e.g., Niazi et al. 2025; Kahil et al. 2025).”*

- The sudden discussion of Spain and Portugal in the abstract felt a bit surprising and arbitrary. Maybe rearranging your current text could soften some of the surprise.

The sentence now reads *“Groundwater recharge changes between 2001 and 2006 show plausible patterns that, for example, align with droughts in southern Europe during this period.”*

- “The pressure on groundwater systems intensifies due to the combined effects of population growth, socioeconomic development, agricultural intensification, and climate change, e.g., through a change in groundwater recharge (Taylor et al., 2013; Reinecke et al., 2021).” I find the cited literature to be insufficient. These papers may have referenced these issues or addressed a subset of these issues, but as far as I can tell, the list of papers does not cover the full space. It could be helpful to cite the papers that have addressed these issues holistically, including those that have looked at combined effects of multiple global change factors on groundwater.

We have included additional references that address these issues. The passage now reads (lines 56 ff):

*“The pressure on groundwater systems intensifies due to the combined effects of population growth, socioeconomic development, agricultural intensification (Niazi et al. 2024; Wada et al. 2012), and climate change (Taylor et al., 2013; Gleeson et al., 2020, Cuthbert et al., 2023, Huggins et al., 2023), e.g., through a change in groundwater recharge (Portmann et al., 2013; Hartmann et al. 2017; Reinecke et al., 2021; Berghuijs et al., 2024; Kumar et al. 2025).”*

- Line 71: (1) what is a “large-scale perspective”? (2) You are citing economics as a key issue, which again makes me concerned that your paper actually omits a lot of humans- and economics-focused global groundwater literature. (3) The cited literature is also non-exhaustive.

As the Editor mentioned, many local and regional hydro-economic models include certain groundwater dynamics. For example, there is a global hydro-economic model that has recently been published (Kahil et al., 2025; https://gmd.copernicus.org/articles/18/7987/2025/gmd-18-7987-2025.html), which includes economics, optimization and groundwater. In fact, there is a wealth of literature in hydro-economics and groundwater; however, currently our paper focuses on the physical side of groundwater dynamics, considering interactions with relevant boundary conditions like socio-economics, agriculture, ecosystem and environment, all of which needs in depth attention. To keep our original focus, we would respectfully acknowledge the groundwater interactions with socio-economy, however, a detailed literature review on economics might be out of the scope in our current manuscript, and to avoid potential ambiguity in our focus. We have further included more literature regarding groundwater studies that have looked at groundwater and socio-economy (see also our responses above).

The first line of the paragraph now reads (lines 65 ff*): “Understanding the impacts of climate change and the globalized socio-economy on groundwater systems (Rodella et al., 2023; Gisser et al., 1980) requires a large-scale perspective that extends from continental to global scales (Haqiqi et al., 2023; Konar et al., 2013; Dalin et al., 2017, Gleeson et al., 2021).”*

- Line 79: This statement seems like it could benefit from further literature support: "While large-scale climate-groundwater interactions are starting to become understood (Cuthbert et al., 79 2019), current global water and climate models may not always capture these feedbacks as most either do not consider groundwater at all or only include a simplified storage bucket, limiting our understanding of how climate change will affect the water cycle as a whole."

We agree and we have added an additional reference. It now reads (lines 74 ff): "*While large-scale climate-groundwater interactions are starting to become understood (Cuthbert et al., 2019), current global water and climate models may not always capture these feedbacks as most either do not consider groundwater at all or only include a simplified storage bucket, limiting our understanding of how climate change will affect the water cycle as a whole (Gleeson et al., 2021, Condon et al, 2021).*"

- I suggest better explaining what the role of the global water sector in isimip is, so readers can better understand the full picture and the implications of a groundwater sector.

We have extended the explanation in section 2 with the following statement(lines 162 ff):

"*Collaboration (and perhaps integration) with sectors like the Global Water sector is possible and desirable in the future. The global water sector focuses on using the ISIMIP protocol to drive a diverse set of global water models (including hydrological and land surface models; Reinecke et al. 2025b) and to produce output variables that capture diverse hydrologic processes, such as discharge, as well as human water use.*"

Results

1. In Figure 4, I found it confusing that you say you'll focus on green and orange—and then orange is called 'focus'.

Thank you for spotting this! We agree that this is confusing. We have changed the figure to now read "under development" for the orange lines.

[Figure]

*Figure 4: The Groundwater sector provides the potential for multiple interlinkages between different sectors within ISIMIP. In the coming years, we will focus on links to three sectors (green and orange): Water (global), Water (regional), and Water Quality. Other cross-sectoral linkages between non-Groundwater sectors (i.e., linkages between the outer circle) are not shown. Sectors that are currently under development or have not yet have data or outputs that could be shared or used for cross-sectoral assessments are shown in gray. Interactions between sectors are annotated with example processes, key variables, or datasets that can be shared between sectors.*

Vision

- You note that "In summary, the ISIMIP Groundwater sector aims to enhance our understanding of the impacts of climate change and direct human impacts on groundwater and a range of related sectors." It's certainly possible I missed it, but I don't see that this is actually supported/made clear in the paper. What set of experiments will

be run that will isolate the impacts of humans on groundwater? What dimensions of human impacts will be looked at, and how? I think this probably just relates to my earlier comment that I'm struggling with wanting more details about what experiments will be done, whereas doing so isn't necessarily the focus of this paper.

Thank you for pointing this out. We already state which experiments will be carried out but possibly not clearly enough. The overall ISIMIP protocol has already defined the experiments, which we also describe in the manuscript. We state in section 2 "The simulation rounds consist of two main components: ISIMIP3a and ISIMIP3b, each serving distinct purposes. ISIMIP3a focuses on model evaluation and the attribution of observed climate impacts, covering the historical period up to 2021. It utilizes observational climate and socioeconomic data and includes a counterfactual "no climate change baseline" using detrended climate data for impact attribution. Additionally, ISIMIP3a includes sensitivity experiments with high-resolution historical climate forcing and water management sensitivity experiments.

In contrast, ISIMIP3b aims to quantify climate-related risks under various future scenarios, covering pre-industrial, historical, and future projections. ISIMIP3b is divided into three groups: Group I for pre-industrial and historical periods, Group II for future projections with fixed 2015 direct human forcing, and Group III for future projections with changing socioeconomic conditions and representation of adaptation. Despite their differences in focus, time periods, and data sources, both ISIMIP3a and ISIMIP3b require the use of the same impact model version to ensure consistent interpretation of output data, thereby contributing to ISIMIP's overall goal of providing a framework for consistent climate impact data across sectors and scales." And state clearly that we aim to carry out these experiments: "In the short term, the Groundwater sector will focus on the historical period from 1901 to 2019 in ISIMIP3a (https://protocol.isimip.org/#/ISIMIP3a/water_global/groundwater), utilizing climate-related forcing based on observational data (obsclim) and the direct human forcing based on historical data (histsoc)."

To make it more apparent that ISIMIP forms the framework for implementing concrete experiments, we added the following paragraph at the end of section 2(lines 176 ff):

*"Carrying out the ISIMIP experiments in the groundwater sector will yield a new understanding of how these models differ, why they differ, and how they could be improved. These experiments will further help to disentangle the impacts of climate change and water management, specifically through ensemble runs of future scenarios using ISIMIP3 inputs."*

- A lot of what you describe is indeed what other communities (including, but not limited, to other isimip sectors) have found about the value of producing ensembles that enable intercomparison. I suggest citing those other communities' work.

We added a non-exhaustive list with a new sentence to section 6 (lines 371 ff): "*Other intercomparison and impact assessment projects already have been successful in achieving similar goals such as the lake (Golub et al., 2022) or water quality sector (Strokal et al., 2025) in ISIMIP, the CMIP (Eyring et al., 2016a), or the AgMIP for agricultural models (von Lampe et al., 2014)."*

**2 Reviewer 1**

The authors provided thoughtful and comprehensive responses to all reviewer comments. To the extent that is reasonable and practical, the authors revised the paper to address reviewer comments. The revisions (additional text + revised Figure 4) have substantially improved the paper, particularly Sections 4 through 6 where both reviewers identified several areas for improvement. I found that the revisions helped clarify the goals (both near and longer term) of the ISIMIP Groundwater sector and strengthened the motivation for a Groundwater-focused sector within ISIMIP.

After considering the comments of the other reviewer and Editor and reading the revised manuscript I think that some revisions, discussed below, could improve the paper.

We thank the reviewer for taking the time to review our revised manuscript.

Comments:
• Regarding distinguishing the paper's contribution raised by the Editor and Reviewer 2. I agree that some additional revisions to the Abstract and Introduction, as suggested by the Editor, could help make the objective of the paper clearer (what it is – motivation and vision for the Sector versus what it isn't – detailed protocols or comprehensive results). Relatedly, currently about 40% of the Abstract is devoted to summarizing the preliminary model results from Section 4. Reducing the emphasis on the model results in the Abstract and instead focusing on other aspects of the paper, such as content in Sections 5 and 6 would make the Abstract a more balanced summary of the paper and help distinguish its core contributions. To me, the value of the paper is introducing the global groundwater community and more broadly the global research community who may be interested in groundwater (e.g. the many connections in Figure 4) to this new sector, and I think the paper accomplishes that well. As pointed out by the Editor, some additional clarifications could be help avoid misunderstandings of the scope or intent of the Sector.

Thank you for this suggestion. In addition to implementing the editor suggestions, we also rewrote parts of the abstract to give more emphasis to sections 5 and 6 as suggested.

The abstract now reads:

*"Groundwater serves as a crucial freshwater resource for people and ecosystems, playing a vital role in adapting to climate change. Yet, its availability and dynamics are affected by climate variations, changes in land use, and abstraction. Despite its importance, our understanding of how global change will influence groundwater in the future remains limited. Multi-model ensembles are powerful tools for impact assessments; compared to single-model studies, they provide a more comprehensive understanding of uncertainties and enhance the robustness of projections by capturing a range of possible outcomes. However, to date, no ensemble of groundwater models has been available to assess the impacts of global change. Here, we present the new Groundwater sector within ISIMIP, which combines multiple global, continental, and regional-scale groundwater models. We describe the rationale for the sector, the sectoral output variables that underpinned the modeling protocol, and showcase current*

*model differences and possible future analysis. Currently, eight models are participating in this sector, ranging from gradient-based groundwater models to specialized karst recharge models, each producing up to 19 out of 23 modeling protocol-defined output variables. To showcase the benefits of a joint sector, we utilize available model outputs of the participating models to show the substantial differences in estimating water table depth (global arithmetic mean 6 - 127 m) and groundwater recharge (global arithmetic mean 78 - 228 mm/y), which is consistent with recent studies on the uncertainty of groundwater models, but with distinct spatial patterns. We further outline synergies with 13 of the 17 existing ISIMIP sectors and specifically discuss those with the global water and water quality sectors. Finally, this paper outlines a vision for ensemble-based groundwater studies that can contribute to a better understanding of the impacts of climate change, land use change, and abstractions on the world's largest accessible freshwater store – groundwater."*

• I agree with concerns raised by the Editor regarding data availability meeting GMD standard for reproducible and open science.

We addressed this comment above as a response to the editor.

• Minor comment: the double use of "focus" for Figure 4 that the editor raised could be fixed by rephrasing to "prioritize the green and orange."

Thank you for this suggestion! We chose to revise the figure, which should now be clearer and consistent with the figure description. Please see our response to the editor above.

**3 Reviewer 2**

This paper is a good step toward improving the incorporation of groundwater processes in global and continental scale climate modeling efforts. It also makes a good case for inclusion in ISIMIP a Groundwater Sector.

We thank the reviewer for taking the time to review our revised manuscript.

My comments and edits are all minor and marked directly on the PDF that I am uploading. That document is best viewed as 2 pages per screen.

Summary of my more relevant comments (but even the other comments/edits not listed here should be read by the authors):
P. 5-6. General comments on authors' responses to prior reviewers.

We thank the reviewer for these extensive comments on our original review replies. We especially acknowledge the comments concerning groundwater quality modeling. Regarding comments about comparing apples with oranges, whether models include water use or not, we would like to note that this paper only describes the experimental setup, which is very inclusive. We agree that the specific scientific analysis that follows the model simulations needs to be

more selective of the model output. Some questions may include a more diverse set of models, and others might require a very selective subset of model simulations.

P. 18. Reference to Table 1: Missing from this list is any mention of shallow and deep groundwater phenomena. See Condon et al. 2021 for several important descriptions of why both shallow and deep (e.g., semi-confined and confined systems, where groundwater levels and interplay with surface processes tend to be completely different than the water table) groundwater phenomena and their connections can also be important for adequately representing groundwater storage and impact processes. And let me add -- this interaction between shallow and deep groundwater can be every bit as important as the interaction between the water table and land surface. The latter is by now a traditional focus on this type of modeling, but it ignores deeper phenomena controlling the upward propagation of deep pumping effects. That phenomena cannot be captured by an approach where the sole hydraulic state variable is the water table elevation or depth.

*We agree that the interaction between deep and shallow groundwater can be important. However, Table 1 provides a descriptive overview of the current generation of models participating in the presented sector. Table 2, which the reviewer highlights with the same comment in the attached PDF, on the other hand, lists several outputs currently part of the new ISIMIP sector. Since none of the models simulate deep groundwater, and there is no global dataset that would allow appropriate parameterization, we do not include specific outputs for this sector regarding these processes. This does not mean they are irrelevant. If, at a later stage, model advances make it possible to provide these outputs, we would certainly be open to changing the protocol to include these variables as well. This has already been best practice in other sectors of ISIMIP to adapt to research advances and needs.*

P. 26 Fig. 1: The map would be much more informative if there was more color contrast in the 0 to 5 m range. Suggest using a different color gradation.

It depends on what aspect one would like to highlight. In Reinecke et al. (2024) Fig. 1a https://iopscience.iop.org/article/10.1088/1748-9326/ad8587/meta we did just that. In principle, a sequential color scale is most appropriate for sequential data. If one is mainly interested in shallow water tables, a greater contrast will highlight differences within that range. Yet, here we chose to use a scale that is clearer also to colleagues looking at deeper groundwater.

P. 27 Fig. 2b: The units are clearly wrong. Perhaps it should be "m/yr"? But even that would not look consistent with the ordinate axis units in (a).

Thank you for spotting this! The figure shows the coefficient of variation as also correctly described in the figure text. But the figure itself should not show a unit at all. While the map is correct the legend description was a mistake. We have changed this and the figure description and map are now correct:

[Figure]

***Figure 2:*** *Global groundwater recharge (GWR) in 2001 or at steady-state (only VIC-wur). The simplified boxplot (a) shows the arithmetic model mean as a colored dot and the median as a black line. Whiskers indicate the 25th and 75th percentiles, respectively. The global map (b) shows the coefficient of variation of the model ensemble without V2KARST calculated as the ensemble standard deviation divided by the ensemble mean. Models shown are not yet driven by the same meteorological forcing (see also table A1).*

P. 30: It seems to me that the biggest benefit of the higher fidelity regional groundwater models is to provide 'test beds' for how to upscale processes in the global or continental scale groundwater models. The latter will necessarily neglect or upscale processes in the former, and there should be comparison studies exploring how to adequately represent regional phenomema in the global/continental models.

Thank you for highlighting this. We agree. However, we also mainly focus regional hydrological models in this section and not necessarily regional groundwater models. It would be great if regional groundwater models would join this effort as well. Unfortunately, so far the interest of local modeling groups has been limited.

We further changed multiple sentences to improve text clarity based on the comments in the attached PDF. We specifically thank the reviewer for providing multiple editorial improvements which we implemented as suggested.

**References**

Niazi, H., Ferencz, S. B., Graham, N. T., Yoon, J., Wild, T. B., Hejazi, M., Watson, D. J., and Vernon, C. R.: Long-term hydro-economic analysis tool for evaluating global groundwater cost and supply: Superwell v1.1, Geosci. Model Dev., 18, 1737–1767, https://doi.org/10.5194/gmd-18-1737-2025, 2025.